# Building-resolving simulations of anthropogenic and biospheric $CO_2$ in the city of Zurich with GRAMM/GRAL

Dominik Brunner<sup>1</sup>, Ivo Suter<sup>1</sup>, Leonie Bernet<sup>1</sup>, Lionel Constantin<sup>1</sup>, Stuart K. Grange<sup>1,\*</sup>, Pascal Rubli<sup>1</sup>, Junwei Li<sup>2</sup>, Jia Chen<sup>2</sup>, Alessandro Bigi<sup>3</sup>, and Lukas Emmenegger<sup>1</sup>

**Correspondence:** Dominik Brunner (dominik.brunner@empa.ch)

Abstract. Urban areas are significant contributors to global  $CO_2$  emissions, requiring detailed monitoring to support climate neutrality goals. This study presents a high-resolution modeling framework using GRAMM/GRAL, adapted for simulating atmospheric  $CO_2$  concentrations from anthropogenic and biospheric sources and sinks in Zurich, Switzerland. The framework resolves atmospheric concentrations at the building scale, and it employs a detailed inventory of anthropogenic emissions as well as biospheric fluxes, which were calculated using the Vegetation Photosynthesis and Respiration Model (VPRM). Instead of simulating the full dynamics of meteorology and atmospheric transport, the dispersion of  $CO_2$  is precomputed for more than 1000 static weather situations, from which the best match is selected for any point in time based on the simulated and measured meteorology in and around the city. In this way, time series over multiple years can be produced with minimal computational cost. Measurements from a dense network of mid-cost  $CO_2$  sensors are used to validate the model, demonstrating its capability to capture spatial and temporal  $CO_2$  variability. Applications to other cities are discussed, emphasizing the need for high-quality input data and tailored solutions for diverse urban environments. The work contributes to advancing urban  $CO_2$  monitoring strategies and their integration with policy frameworks.

## 1 Introduction

Cities play a crucial role in tackling the challenge of climate change as they are responsible for close to 70% of global  $CO_2$  emissions (Seto et al., 2014). Cities around the world have recognized their responsibility and set ambitious goals to reduce emissions in support of the Paris Climate Agreement. The city of Zurich, for example, aims to become climate neutral by 2040 following the public vote on the 'Net Zero 2040 Climate Protection Target'.

The Enhanced Transparency Framework of the Paris Agreement provides guidelines for countries on how to communicate their targets and collect and report their emissions. Due to these standardized procedures and the availability of comprehensive socioeconomic data, emission inventories are often quite accurate on a national scale. This is particularly true for CO<sub>2</sub> emissions, which are usually dominated by fossil fuel consumption, for which accurate national statistics exist. At the city level, in contrast, reliable statistical data are often lacking and domain boundaries are less clearly defined, for example, due

<sup>&</sup>lt;sup>1</sup>Empa, Swiss Federal Laboratories for Materials Science and Technology, Dübendorf, Switzerland

<sup>&</sup>lt;sup>2</sup>Electrical and Computer Engineering, Technische Unversität München, Munich, Germany

<sup>&</sup>lt;sup>3</sup>Dipartimento di Ingegneria "Enzo Ferrari", Universita degli Studi di Modena e Reggio Emilia, Modena, Italy

<sup>\*</sup>now at Queensland University of Technology, Brisbane, Australia

to commuter traffic and the consumption of energy produced outside the city. Although the Global Protocol for Community-Scale Greenhouse Inventories (GPC) offers a widely accepted emission reporting framework (Fong et al., 2014), most cities still follow their own procedures, which vary greatly in terms of detail and accuracy (Gurney et al., 2021; Ahn et al., 2023).

To track progress towards the reduction targets, consistent, reliable, and timely emission information is required. National and city-scale emission inventories are traditionally compiled "bottom-up" from statistical activity data and emission factors, or from fuel consumption data as in the case of  $CO_2$ . Depending on the quality and completeness of the input data and the resources available to process these data, the inventories may be associated with considerable uncertainty and are often delayed by several years. In a recent study, Gurney et al. (2021) compared Scope 1 greenhouse gas emissions reported by 48 cities in the U.S. with independent estimates from the Vulcan  $CO_2$  emission data product. They found relative differences between -145.5% and 63.5%, which highlights the large uncertainty in self-reported city inventories.

It is therefore desirable to verify these inventories with independent data and to complement the estimates with more timely information. Such independent estimates can be provided by "top-down" methods that combine measurements of atmospheric concentrations with atmospheric transport simulations (Bergamaschi et al., 2018). Emissions from a city produce a gradient in atmospheric concentrations between regions upstream and downstream, which can be measured by a measurement network with stations suitably placed to capture these gradients. An atmospheric transport model is required to provide the quantitative link between emissions and measured concentrations. Starting from an a priori estimate of the emissions based on a bottom-up inventory, observations and the transport model information are then combined in a mathematical inversion. This process generates an a posteriori estimate that brings the simulated concentrations closer to the observations.

Top-down methods have been demonstrated for a number of cities including Indianapolis (Turnbull et al., 2018; Lauvaux et al., 2020), Los Angeles (Brioude et al., 2013), Boston (Sargent et al., 2018) and Paris (Nalini et al., 2022; Staufer et al., 2016). However, important questions remain regarding the suitability of different observation and modeling strategies. Cities vary greatly in terms of size, structure and topography and in terms of their natural, economic and societal environment. It is therefore unlikely that a single monitoring strategy optimally fits all cities.

Against this background, the EU Horizon 2020 Project ICOS-Cities / PAUL (Pilot Applications in Urban Landscapes) was initiated to evaluate different monitoring strategies in three European cities varying in size and complexity, namely Paris, Munich and Zurich. The study presented here focuses on Zurich, which is the smallest of the three cities but the largest agglomeration in Switzerland with 447,082 inhabitants (31 Dec 2023) within the city boundaries and an additional population of approximately 1,000,000 in the surrounding agglomerations. The city has a complex topography (see Fig. 1) with the Albis mountain ridge in the west (tallest elevation Üetliberg 870 m a.s.l.) and a series of hills further in the east aligned along a northwest to southeast axis with Hönggerberg (541 m a.s.l.), Käferberg (571 m a.s.l.), Zürichberg (676 m a.s.l.) and Adlisberg (701 m a.s.l.). The main part of the city is in the Limmat Valley (valley floor at about 400 m a.s.l.) between Albis and this line of hills. Additional districts are in the Glatt Valley on the eastern side of the hills.

In order to account for the complex landscape and topography of Zurich, a dense network of low- and mid-cost  $CO_2$  sensors has been set up providing measurements at street level and above rooftop. The network of mid-cost sensors, namely ZiCOS-M, is described in detail in Grange et al. (2024). The city network is complemented by three background sites, of which two

**Figure 1.** Model domains and sensor locations overlaid over a map of the city of Zurich. The red dashed line is the GRAMM domain, the red solid line the GRAL domain. The black solid line denotes the political boundaries of the city. Different types of sensors are shown with different symbols as indicated by the legend. Background map: © OpenStreetMap contributors 2024. Distributed under the Open Data Commons Open Database License (ODbL) v1.0.

are equipped with high-precision Picarro instruments and one with a mid-cost sensor. To enable an alternative method for quantifying the  $CO_2$  emissions of the city, an Eddy covariance system for direct  $CO_2$  flux measurements was installed on a 16.5 m mast on top of one of the tallest (95.3 m) buildings in the city. These measurements are presented in Hilland et al. (2025) and were not used in this study.

Networks of low- to mid-cost CO<sub>2</sub> sensors have already been deployed in other cities. A prominent example is the Berkeley Atmospheric CO<sub>2</sub> Observation Network (BEACO<sub>2</sub>N), a dense network of 35 nodes of CO<sub>2</sub> and air pollution sensors in the San Francisco Bay area (Shusterman et al., 2016), which was recently extended to other cities including Los Angeles (Kim et al., 2025), Glasgow, Providence and Heidelberg. Other examples are the Beijing–Tianjin–Hebei (JJJ) carbon monitoring system with 134 CO<sub>2</sub> sensor sites (Han et al., 2024) and a mid-cost CO<sub>2</sub> sensor network with eight sites in the city of Paris (Lian et al., 2024). In an inverse modeling study utilizing a mesoscale Lagrangian particle dispersion model, Turner et al. (2020) showed that the BEACO<sub>2</sub>N network successfully captured the reduction of CO<sub>2</sub> emissions during the COVID-19 pandemic.

To constrain the  $CO_2$  emissions of the city of Zurich, the measurements from the ZiCOS-M network will be used in an inverse modeling framework together with high-resolution atmospheric transport models. Here, we describe one of these model systems, GRAMM/GRAL, which was previously used for air pollution simulations in the city of Zurich (Berchet et al., 2017a, b) and for optimizing the design of a  $CO_2$  measurement network in the city of Heidelberg using an observing system simulation experiment (OSSE) (Vardag and Maiwald, 2024). The model operates at a high horizontal, building-resolving resolution, which is required to capture the situation at the low-cost sensor sites at street level, but it may also be an advantage to represent the  $CO_2$  concentrations at rooftop level. Due to computational constraints, the horizontal resolution was limited to 10 m, which is sufficient to capture the flow in wide streets and open spaces such as parks, squares and river, but it poses limitations for resolving the flow in most street canyons in Zurich, which are no wider than a few tens of meters. Despite these limitations, the resolution is orders of magnitude better compared to atmospheric transport models used in previous urban  $CO_2$  studies, which typically operated at resolutions of one kilometer or coarser (Staufer et al., 2016; Feng et al., 2016; Turner et al., 2020; Lauvaux et al., 2020; Nalini et al., 2022). Computational fluid dynamics models such as OpenFoam (Kubilay et al., 2018) and PALM (Maronga et al., 2020) have also been run over cities at building-resolving scale down to sub-meter resolution, but these simulations were limited to individual districts or short time periods due to the excessive computational costs.

To take full advantage of GRAMM/GRAL, the  $CO_2$  simulations need to be fed by emissions from an inventory of comparable resolution. The city of Zurich has developed a detailed GIS (Geographical Information System) inventory for air pollutants and  $CO_2$ , which describes the emissions of 65 individual sectors as point, line or area sources. This inventory is used as input for GRAMM/GRAL to compute the distribution of anthropogenic  $CO_2$  within the city at hourly resolution.

Urban  $CO_2$  concentrations are not only affected by anthropogenic emissions, but also by the exchange of  $CO_2$  with the biosphere. The impact of vegetation on urban  $CO_2$  can be in a similar range as the anthropogenic signal, especially during the growing season when plants take up  $CO_2$  by photosynthesis (Sargent et al., 2018). In order to isolate the anthropogenic signal, it is therefore imperative to also simulate the influence of vegetation.

The aim of this study is to provide a detailed description of the GRAMM/GRAL model system together with the input data required for the simulation of anthropogenic and biospheric  $CO_2$  and to demonstrate its performance by comparing simulated and measures  $CO_2$  concentrations. This study forms the basis for subsequent work on the assimilation of the observations in an inverse modeling framework with GRAMM/GRAL.

Sect. 2 discusses the model set-up, the measurement network, the emission inventories, as well as the representation of CO2 contributions from human respiration and vegetation. Sect. 3 describes the modeling chain applied to generate time series of

 $CO_2$  maps and station concentrations. Finally, Sect. 4 presents the performance of the model in terms of spatial distribution and temporal variability compared with in-situ measurements.

### 2 Data and methods



#### 100 2.1 Low- and mid-cost sensor network

The sensor network consists of 61 sites with paired low-cost  $CO_2$  sensors and 22 sites with mid-cost sensors (see Fig. 1). The total number of measurement locations is 77, since 6 of the 22 mid-cost sites are also equipped with low-cost sensors to better characterize their performance. The low-cost sensors are mostly mounted on lamp poles close to the ground (typically 3 - 5 m a.g.l.) and often adjacent to roads. The mid-cost sensors, in contrast, are primarily used for measurements at telecommunication antennas of the company Swisscom above rooftops (15 out of 22). The low-cost sensors are Senseair LP8 NDIR (Non-dispersive Infrared) sensors, which were integrated into a small housing with battery power and Low-Power Network (LoRaWan) communication by the commercial partner Decentlab. Their precision of about 10 ppm is sufficient to measure increments produced by local emissions such as nearby streets as well as accumulation of  $CO_2$  in stable nocturnal boundary layers, but not to measure city-scale gradients. More details on the low-cost sensors and their calibration is presented in Müller et al. (2020). The calibration strategy has recently been revised to better incorporate information from the mid-cost sensors, which will be described in a forthcoming publication.

The mid-cost sensors are based on the same NDIR technique but with important differences: They are more sensitive, temperature-stabilized, mostly mounted inside air-conditioned rooms, and ambient air is actively drawn in by a pump, often through several tens of meters of sampling line from the inlet above rooftop. Furthermore, the sensors are calibrated daily by supplying air from two calibration gas cylinders with approximately 400 ppm and 600 ppm of CO<sub>2</sub> referenced to the WMO X2019 scale. Three different brands of mid-cost sensors are used: Senseair HPP, Vaisala GMP 343, and LI-COR 850, all integrated into the same housing assembled by Decentlab and equipped with LoRaWan communication for automated data transfer. Before deployment in the field, all low- and mid-cost sensors were characterized during outdoor measurements in parallel with a high-precision instrument (Picarro) at Empa, Dübendorf (see Fig. 1). The mid-cost sensor network and its performance are described in detail in Grange et al. (2024).

The city network was completed in early July 2022 and operated fully until September 2024, followed by a scale-down phase to optimize resources. Although the rooftop locations were carefully selected to be well above roof level and away from local sources, contamination by nearby chimneys and ventilation shafts could not always be avoided. In November 2022, many of the mid-cost sites were therefore equipped with additional meteorological sensors (wind, temperature, humidity, pressure) to better characterize the influence of nearby sources as a function of wind conditions and to flag contaminated observations. These sites, denoted with red circles in Fig. 1, provide invaluable wind information for the GRAMM/GRAL simulations.

Three sites at distances of 13 km to 33 km from the city center are used to characterize background levels of CO<sub>2</sub>. Two of the sites (Beromünster, Lägern-Hochwacht) are equipped with high-precision Picarro G2401 instruments (Oney et al., 2015). They are located outside the map presented in Fig. 1 towards the southwest (Beromünster) and northwest (Lägern), respectively.

The background site Breite-Birchwil is located to the northeast of Zurich (blue circle in the upper right part of Fig. 1) and is equipped with a LI-COR 850 mid-cost sensor.

### 2.2 GRAMM/GRAL model setup





The model system GRAMM/GRAL was developed at the Graz University of Technology and consists of the non-hydrostatic weather model GRAMM and the computational fluid dynamics and Lagrangian particle dispersion model GRAL, both at version v19.01. The dispersion of CO<sub>2</sub> from vegetation was computed later using version v23.11 of GRAL. The performance of GRAMM/GAL in terms of representing meso- and microscale flows and pollutant dispersion has been evaluated extensively in previous studies. Using a similar setup as in our study, May et al. (2024) investigated how well the flow in the complex topography of the city of Heidelberg, Germany, is represented by the model. They found very good agreement at 11 out of 15 measurement sites according to the performance criteria for mesoscale air quality models formulated by the European Environment Agency (Agency, 2011). At all 15 sites, the performance was within the extended limits proposed by Oettl and Veratti (2021) for the more challenging conditions in complex terrain. Furthermore, Oettl and Veratti (2021) showed that winds simulated by GRAMM in Alpine topography are at least as accurate as those simulated by the numerical weather prediction model WRF at comparable resolution. The high quality of tracer dispersion simulated by the GRAL model has been demonstrated in several studies comparing model results with wind tunnel and tracer release experiments (Oettl, 2015) and with air pollutant measurements in street canyons (Oettl and Uhrner, 2011) and across a whole city (Berchet et al., 2017b).

We largely followed the setup presented by Berchet et al. (2017b) for the simulation of nitrogen oxides over Zurich. In particular, we applied the same catalogue approach, which allows simulating extended periods of time at affordable computational cost. Instead of simulating the dynamic evolution over a given period of time, the flow and  $CO_2$  concentration fields are computed for a fixed set of weather situations representing different synoptic forcings (large-scale wind speed and direction) and atmospheric stabilities. For each situation, the dispersion of  $CO_2$  is integrated until a steady-state concentration field is reached. It is assumed that the meteorological conditions and corresponding  $CO_2$  dispersion of any hour of a year can be represented by a static situation from the catalogue. The most appropriate situation is selected by comparing the simulated flow fields with actually measured meteorological conditions (see Sect. 2.7). This approach allows long time series to be constructed at minimal computational cost.

The catalogue consist of 1120 predefined weather situations, which are combinations of 40 wind directions (40 directional sectors of 9°), 7 wind speed classes (0.25, 0.75, 1.5, 2.5, 4, 5.5, 7 m/s) and 7 Pasquill-Gifford-Turner stability classes (Turner, 1964) with classes A (most unstable) to G (most stable). Note that a large number of combinations, e.g. high stability and strong winds, are meteorologically not meaningful and were therefore excluded.

The generation of the catalogue involves three different types of simulations. First, the mesoscale flow over a domain of 30 km x 30 km centered over Zurich is computed with GRAMM at a horizontal resolution of 100 m x 100 m (see Fig. 1). The vertical resolution is 10 m near the ground and stretched with increasing height with a stretching factor of 1.2 up to around 2700 m above ground (22 levels in total). GRAMM accounts for the influence of the topography and land use (roughness, albedo, emissivity, soil moisture and heat capacity) on the mesoscale flow. It was specifically developed for simulations in

steep, mountainous terrain, where many other mesoscale models become unstable (?). All inputs required for the simulations with GRAMM or GRAL, including anthropogenic emissions and inputs required for the computation of biospheric fluxes, are listed in Table 1. Most data sets are static with update cycles between one and five years. When available, the reference year is included in the data set description. The two Sentinel-2 satellites together provide global coverage in five days. Europe is covered more frequently, but clouds lead to irregular sampling and corresponding data gaps of up to a few weeks, especially in winter.

In GRAMM, vertical profiles of temperature and humidity as well as solar azimuth are prescribed according to the stability class. For stable conditions, for example, no solar radiation is assumed, which favors the formation of cold air drainage flows down the hills surrounding the city. An example of the flow at the second model level (centered at 16 m above surface) for a westerly forcing of 2.5 m s<sup>-1</sup> is presented in Fig. 2 for both a stable and an unstable situation. In both situations, winds are strongest on top of the hills and lower in the city center. However, there are also major differences. In the unstable situation, the flow is more strongly distorted due to the presence of thermally-induced wind systems such as up-slope winds along the hills and a lake breeze along the shores of Lake Zurich.






The second simulation is a GRAL flow simulation for a smaller domain of 13 km x 13 km at much higher horizontal (10 m x 10 m) and vertical resolution (2 m) in order to resolve the flow around buildings in the city. We used the more accurate but also more expensive option of GRAL to solve the flow prognostically. In this case, GRAL solves the Reynolds averaged Navier-Stokes (RANS) equations neglecting Coriolis and buoyancy forces and simulates turbulent mixing using a mixing-length approach as default option. Obstacles such as buildings are represented as square blocks, and the topography is represented as a sequence of 2 m tall steps. It should be noted that even at such a high resolution, the flow in narrow streets and backyards is not well represented, which limits the comparison with corresponding observations. The GRAL simulation is forced at its lateral (inflow) boundaries by the GRAMM flow simulated in step 1. GRAL uses the same topography without interpolation.

The third simulation is a  $CO_2$  dispersion simulation with a Lagrangian particle dispersion model, again performed by GRAL. Virtual particles carrying a fixed mass of  $CO_2$  are released from emission sources at a rate that is proportional to the source strength. They are then transported forward by the flow computed in the previous step and by stochastic dispersion depending on the stability class. Concentrations are computed by summing up the masses of all particles in a given volume. For further details on the GRAMM and GRAL models we refer to ? and the extensive documentation published online by the Graz University of Technology (2023).

Concentrations were written out at  $10 \text{ m} \times 10 \text{ m}$  horizontal resolution for 9 vertical layers at 2, 5, 10, 20, 30, 45, 60, 75 and 100 m above ground. In order to determine the contribution of different emission sectors and biospheric fluxes,  $CO_2$  fields were computed for 24 different source groups individually, of which 17 groups represent anthropogenic sources, 5 represent biospheric contributions and 2 human respiration (see Table ?? in the Supplement).

Since these concentration fields are representative for an annual average emission rate, they need to be scaled by the ratio of actual emissions to annual average emissions to compute the  $CO_2$  distributions for any given hour of the year. Because the temporal profiles are source-specific, the scaling needs to be applied for each source sector separately.

Table 1. Input data for the GRAMM and GRAL simulations.

| Purpose           | Description              | Resol.         | Source                     | Link                                                |  |  |
|-------------------|--------------------------|----------------|----------------------------|-----------------------------------------------------|--|--|
|                   | GRAMM                    | ſ              |                            |                                                     |  |  |
| Orography         | ASTER global digital     | 30 m           | NASA JPL                   | https://search.earthdata.nasa.gov                   |  |  |
|                   | elevation model V003     |                |                            |                                                     |  |  |
| Land use          | CORINE 2018              | 100 m          | Copernicus Land            | https://land.copernicus.eu/en/products/             |  |  |
|                   |                          |                | Monitoring Service         | corine-land-cover/clc2018                           |  |  |
|                   | GRAL incl. Anthropog     | enic Emissi    | ons                        |                                                     |  |  |
| 3D buildings      | 3D block model           | 0.5 m          | City of Zurich, Geomatik + | https://opendata.swiss/de/dataset/                  |  |  |
|                   | (LoD 1)                  |                | Vermessung (GEOZ)          | bauten-blockmodell-inkl-3d                          |  |  |
| Emissions in city | MapLuft 2020             | $vector^{(1)}$ | City of Zurich, Umwelt und | https://www.stadt-zuerich.ch/geodaten/download/     |  |  |
|                   |                          |                | Gesundheitsschutz (UGZ)    | Emissionskataster_Luftschadstoffe_und_Treibhausgase |  |  |
| Emissions outside | Swiss national inventory | 100 m          | BAFU/Meteotest/INFRAS      | not publicly available                              |  |  |
|                   | 2020 in rasterized form  |                |                            |                                                     |  |  |
| Human respiration | Residents and employees  | $zone^{(2)}$   | Canton of Zurich,          | https://www.geolion.zh.ch/geodatensatz/             |  |  |
|                   | Quartieranalyse          |                | Statistical Office         | 3233                                                |  |  |
|                   | VPRM Biospheri           | ic Fluxes      |                            |                                                     |  |  |
| Vegetation cover  | Amtliche Vermessung 2024 | vector(3)      | Canton of Zurich,          | https://geolion.zh.ch/geodatenprodukt/              |  |  |
|                   |                          |                | Statistical Office         | show?gdpnummer=10016                                |  |  |
| Cropland          | Urban Atlas 2018         | 25 m           | Copernicus Land            | https://land.copernicus.eu/en/                      |  |  |
|                   |                          |                | Monitoring Service         | products/urban-atlas/urban-atlas-2018               |  |  |
| Vegetation height | Vegetationshöhenmodell   | 1 m            | WSL, National              | https://opendata.swiss/de/dataset/                  |  |  |
|                   | NFI 2021                 |                | Forest Inventory           | vegetationshohenmodell-lfi                          |  |  |
| Tree type         | Waldmischungsgrad        | 10 m           | WSL, National              | https://opendata.swiss/de/dataset/                  |  |  |
|                   | NFI 2021                 |                | Forest Inventory           | waldmischungsgrad-lfi                               |  |  |
| Satellite indices | Sentinel-2               | 10 m           | Copernicus Land            | https://dataspace.copernicus.eu/explore-data/       |  |  |
|                   |                          |                | Monitoring Service         | data-collections/sentinel-data/sentinel-2           |  |  |

<sup>(1)</sup> Point, line and area sources in GIS format.

As also described by Berchet et al. (2017b), the simulated  $CO_2$  concentration c for any given hour h and location  $\mathbf{x}$  is thus obtained as

$$c(h, \mathbf{x}) = c_{bg}(h) + \sum_{i \in sectors} \tau_i(h) \cdot c_i(\eta(h), \mathbf{x}),$$
 (1)

<sup>(2)</sup> A zone is a connected group of buildings bounded by streets, rivers etc. A zone has a minimum size of 100 m<sup>2</sup>.

<sup>(3)</sup> Vector data set with an accuracy of 3 cm.

**Figure 2.** Example of GRAMM winds for (a) a stable (class F) and (b) an unstable (class B) situation, both forced at the lateral boundaries with a westerly flow of 2.5 m/s. The figure does not show the whole GRAMM domain but only a zoom into the area covered by GRAL. Background map: © OpenStreetMap contributors 2024. Distributed under the Open Data Commons Open Database License (ODbL) v1.0.

where  $\tau_i(h)$  is the temporal scaling factor for source sector i,  $c_i(\eta(h),\mathbf{x})$  is the  $\mathrm{CO}_2$  concentration for catalogue entry  $\eta(h)$  at location  $\mathbf{x}$ , and  $c_{bg}(h)$  is the background concentration. All concentrations are reported here and in the following as dry air mole fractions in units of ppm ( $\mu$ mol/mol), which are calculated from the simulated  $\mathrm{CO}_2$  mass densities ( $\mathrm{kg}\,\mathrm{m}^{-3}$ ) by division with the density of dry air and multiplication with the ratio of the molar masses of dry air (28.96 g mol<sup>-1</sup>) and  $\mathrm{CO}_2$  (44.01 g mol<sup>-1</sup>). The temporal profiles applied for the different sectors and biospheric fluxes are described in Sect. 2.6. The procedure for identifying the catalogue entry  $\eta(h)$  best matching the current hour h is presented in Sect. 2.7. The estimation of a suitable background  $c_{bg}(h)$  is described in Sect. 2.8.

### 2.3 Anthropogenic emissions


The city of Zurich has developed a detailed inventory of point, line and area sources, called MapLuft. The inventory is currently available for the years 2010, 2015, 2020 and 2022. It covers 9 air pollutants (PM10, PM2.5, NO<sub>x</sub>, CO, SO<sub>2</sub>, NH<sub>3</sub>, VOCs, soot, benzene) and 3 greenhouse gases (CO<sub>2</sub>, CH<sub>4</sub>, N<sub>2</sub>O) divided into 65 different source sectors (see Supplement Table ??). Here,

**Figure 3.** Anthropogenic CO<sub>2</sub> emissions from Zurich's MapLuft inventory in 2022 at (a) 100 m x 100 m and (b) 10 m x 10 m resolution (right). Panel (a) includes emissions from the gridded Swiss CO<sub>2</sub> inventory outside the city boundaries. Panel (b) also displays the 10 largest point sources as black/red circles with areas proportional to the emissions.

we only use the inventory for  $CO_2$ . Maps of the total  $CO_2$  emissions from all sectors are presented in Fig. 3 in two different resolutions. The left panel shows emissions at 100 m resolution and includes emissions outside the city boundaries taken from the Swiss  $CO_2$  inventory. The right panel shows the MapLuft emissions within the city boundaries at 10 m resolution with the ten largest point sources overlaid as circles with sizes proportional to emission strength. By far the largest point source accounting for 244.6 kt or 19.5% of total  $CO_2$  emissions in 2022 is the waste-to-energy plant Hagenholz in the northeastern corner of the city. Note that the emissions were provided to GRAL directly as point, line or area sources rather than in a gridded format. Emissions from heating systems, for example, were released above roof level at the exact locations and altitudes of more than 24,000 chimneys in the city.

The processing of the GIS-inventory to the input format required by GRAL was performed using the python tool emiproc (Lionel et al., 2025). In order to save computation time, the 65 source sectors from MapLuft were lumped into 10 broader groups before simulating them in GRAL (see Supplement Table ??). The grouping was based on two considerations: (a) Sectors should have similar temporal variability such that their activity can be represented by a single temporal profile. (b) Fossil and non-fossil sources should be in separate groups whenever possible. Three additional groups (IDs 71, 72, 74) were simulated to separate oil from other fossil heating systems and to distinguish between the two waste-to-energy power plants. Isolating oil heating systems was considered useful as their emissions are declining rapidly due to their replacement by gas and district heating. Sector 74 represents only the emissions from the waste-to-energy plant Hagenholz, which allows separating it from a similar plant inside the city center that was turned off in summer 2021, before the start of the measurements.


Table 2. Annual total anthropogenic emissions (excluding human respiration) in the city of Zurich per GNFR category.

| Sector                          | CO <sub>2</sub> fos (kt) | CO <sub>2</sub> bio (kt) | CO <sub>2</sub> fos (%) | CO <sub>2</sub> bio (%) |  |
|---------------------------------|--------------------------|--------------------------|-------------------------|-------------------------|--|
| A - Public Power                | 134.7                    | 127.2                    | 10.72                   | 10.13                   |  |
| B - Industry                    | 10                       | 0                        | 0.80                    | 0.00                    |  |
| C - Other stationary combustion | 554.2                    | 21.5                     | 44.12                   | 1.71                    |  |
| D - Fugitives                   | 0                        | 0                        | 0.00                    | 0.00                    |  |
| E - Solvents & product use      | 0                        | 0                        | 0.00                    | 0.00                    |  |
| F - Road transport              | 354.4                    | 8.7                      | 28.21                   | 0.69                    |  |
| G - Shipping                    | 0.9                      | 0                        | 0.07                    | 0.00                    |  |
| H - Aviation                    | 0                        | 0                        | 0.00                    | 0.00                    |  |
| I - Off Road                    | 16                       | 0                        | 1.27                    | 0.00                    |  |
| J - Waste                       | 27.8                     | 0                        | 2.21                    | 0.00                    |  |
| K - AgriLivestock               | 0                        | 0                        | 0.00                    | 0.00                    |  |
| L - AgriOther                   | 0                        | 0                        | 0.00                    | 0.00                    |  |
| R - Other                       | 0.1                      | 0.7                      | 0.01                    | 0.06                    |  |
| Total                           | 1098.1                   | 158.1                    | 87.41                   | 12.59                   |  |

Four more sectors (IDs 34-37) were added to represent emissions within the GRAL domain but outside the city boundaries.

These were taken from the Swiss-wide Meteotest inventory available on a 100 m x 100 m grid. All data was provided in the Swiss coordinate system LV95, which is the system used for almost all geo-referenced data in Switzerland.

An overview of the absolute and relative contribution of different source sectors is presented in Table 2. For easier comparison with other inventories, the sources were grouped here according to the GNFR (Gridded Nomenclature For Reporting) classification. The percentage numbers refer to the total fossil + biogenic CO<sub>2</sub> emissions of 1256.2 kt in 2022. Biogenic sources made up a non-negligible fraction of 12.6% primarily due to the burning of waste in the Hagenholz plant, for which the study of Mohn et al. (2012) estimated a share of 52% to be non-fossil based on radiocarbon analysis. Other relevant biogenic sources were residential wood burning and biofuels in traffic. The three largest CO<sub>2</sub> source sectors were stationary combustion/heating (45.8%), traffic (28.9%) and public power (20.8%, dominated by Hagenholz).

### 2.4 Biospheric fluxes


The exchange of CO<sub>2</sub> with vegetation was simulated with the Vegetation Photosynthesis and Respiration Model (VPRM) (Mahadevan et al., 2008). VPRM parametrizes gross photosynthetic production (GPP) and respiration (RE) for different vegetation types using a simple set of equations with minimal parameters. The model requires inputs such as air temperature, solar radiation, and satellite-based indices to capture vegetation dynamics and soil water availability.

Although a version of VPRM tailored to urban environments is available (Hardiman et al., 2017), we used the original formulation of Mahadevan et al. (2008) but with updated parameters for the five vegetation types present in the city, i.e. deciduous

trees, evergreen trees, mixed forest, grass and cropland (Supplement Table ??). The parameters were generated following the method of Glauch et al. (2025), and are identical to those presented in Stagakis et al. (2025). A detailed comparison with three other vegetation models and with observations collected in four urban parks in Zurich showed that this new set of parameters produces realistic GPP and RE fluxes (Stagakis et al., 2025).

Standard land cover products such as CORINE do not have sufficient resolution nor properly represent street trees. Therefore, a custom-made land and vegetation cover map (Fig. 4) was produced on the 10 m x 10 m resolution grid of GRAL by combining the following four data sources (see also Table 1):

- Land use cadaster (Amtliche Vermessung) of the Canton of Zurich, the official land registry of the Canton, which served
  as base map.
- The Urban Atlas 2018 from the Copernicus Land Monitoring Service was used to distinguish cropland from grassland,
   which was not possible with the land use cadaster alone.
  - The Vegetation Height Model from the Swiss National Forest Inventory was used to identify trees, especially trees in street canyons and parks. This data set is based on a combination of stereo aerial images and Lidar observations.
  - Forest Mixture (Waldmischungsgrad) from the Swiss National Forest Inventory to distinguish deciduous from evergreen trees. This data set is based on Sentinel-1 and Sentinel-2 satellite observations and a machine learning classification algorithm. A forest mixture ratio above 80% was classified as deciduous, a ratio of 20% to 80% as mixed, and less than 20% as evergreen. Note however that a large majority of pixels classified as mixed inside the city are actually deciduous trees.

From each 10 m x 10 m pixel covered by a given vegetation type,  $CO_2$  was released in GRAL at a fixed rate of  $1.0 \,\mathrm{kg}\,\mathrm{h}^{-1}$  to compute the  $CO_2$  distribution expected from a unit release of  $CO_2$  from all pixels covered by that vegetation (e.g. grass). For any given hour, this distribution was then scaled by the ratio of the actual photosynthetic uptake and respiration of  $CO_2$  computed with VPRM to the unit flux. Further details are provided in Sect. 2.6.2.

## 2.5 Human respiration



Human respiration is a non-negligible source of  $CO_2$  in cities (Ciais et al., 2020). Based on the number of residents and employees from the Quartieranalyse of the Canton of Zurich (see Table 1) and an average emission rate of 1 kg  $CO_2$  per person per day, total emissions in Zurich were estimated to 163 kt yr<sup>-1</sup>, which is 13.0% of the total anthropogenic emissions of 1,256 kt yr<sup>-1</sup> listed in Table 2. A rate of 1 kg per day is close to other estimates, e.g. the global average of 0.903 kg per day suggested by Cai et al. (2022) or the average rate for Chinese citizens of 0.935 kg per day estimated by Wang et al. (2024).

The Quartieranalyse resolves the number of residents and employees at the district level. This information was then distributed over all buildings in the district proportional to building volume to produce the maps shown in Fig. 5. Lacking any information on where and at what altitude exactly  $CO_2$  from human respiration is released to the atmosphere, we assumed that these emissions occur at an altitude of 1 m above any surface within the district including roofs. This assumption is a major

Figure 4. Custom-made land and vegetation cover map for Zurich.

simplification, but a more accurate treatment of human respiration would require a detailed modeling of the person's behavior and mobility data and information of building ventilation systems, which is beyond the scope of the present study.

# 280 2.6 Temporal scaling of emissions and biospheric fluxes

As shown in Eq. 1, the  $CO_2$  concentration simulated for any given hour h is obtained by scaling the concentration fields for each source group i with suitable temporal factors  $\tau_i(h)$ . Note that all emissions from a given source group are varied in the same way across the whole city. Possible temporal differences, for example, in traffic activity between the city center and outskirts cannot be accounted for in this way, unless they are simulated as separate categories. The chosen temporal profiles for anthropogenic emissions, biospheric (VPRM) fluxes and human respiration are described in detail in the following.

## 2.6.1 Anthropogenic emissions and human respiration


Depending on the sector, temporal profiles were either derived from actual activity data or were standard profiles from Kuenen et al. (2014). Traffic (GNFR sector F) was scaled with the ratio of the actual hourly traffic intensity to the annual mean traffic

Figure 5. Number of (a) residents and (b) employees per building block in units of persons per hectare. White color indicates no available information or zero occupancy. Background map: © OpenStreetMap contributors 2024. Distributed under the Open Data Commons Open Database License (ODbL) v1.0.

deduced from 182 traffic counters distributed over the city. Unfortunately, the counters do not distinguish between different modes of transport, such that the same profiles had to be applied to both cars and trucks. The daily mean scaling factors for the period September 2022 to August 2023 and the mean diurnal cycles are shown in Fig. 6. The daily factors reveal a pronounced weekly cycle with reduced traffic on weekends and dips during holidays, especially in the Christmas and summer holiday seasons. The mean diurnal cycle shows a sharp increase in the early morning hours, a gradual growth during the day, and a rapid decrease in the evening hours. Note that local time is UTC+1 in winter and UTC+2 in summer, which leads to some smoothing of the annually averaged profiles when plotted against UTC.



Daily emissions from heating systems (GNFR sector C) were scaled with a heating degree days (HDD) approach, which describes the heating demand depending on the difference between indoor and outdoor temperatures. The daily average heating demand is calculated from a daily mean 2 m outdoor temperature  $T_{2m}$  as follows:

$$HDD(d) = \begin{cases} T_{in} - T_{2m}(d), & \text{if } T_{2m}(d) \le T_{min} \\ 0, & \text{otherwise} \end{cases}, \tag{2}$$

where  $T_{in}$  is the indoor temperature (set to 20°C) and  $T_{min}$  is a threshold temperature below which heating systems are activated (set to 12°C). The activity for a given day is then the ratio of the daily HDD(d) to the annual mean  $\overline{HDD}$ 

$$a_{HDD}(d) = \frac{HDD(d)}{\overline{HDD}}.$$
(3)

**Figure 6.** Temporal scaling factors for traffic and heating. (a) Daily mean factors for the period September 2022 - August 2023. (b) Mean diurnal cycles including the heating demand cycle for hot water production alone.

To obtain an hourly scaling factor, the daily activity is modulated by a diurnal (hour-of-day) activity profile  $p_{HDD}$ . Independent of outdoor temperatures, there is an additional heating demand for hot water production, which also has a diurnal profile  $p_{HW}$ . The final hourly heating activity was computed as



$$a_H(h) = (1 - f_{HW}) \cdot a_{HDD}(D) \cdot p_{HDD}(h) + f_{HW} \cdot p_{HW}(h),$$
 (4)

with  $f_{HW}$  being the fraction of the total heating energy used for hot water production, which was set to 0.233. The diurnal profiles  $p_{HW}$  and  $p_{HDD}$  are provided in the supplementary material. These profiles and the fraction  $f_{HW}$  were derived fromstandard heating profiles defined by the Swiss Society of Engineers and Architects.

For all other sectors, a superposition of three fixed temporal profiles was used representing hour-of-day, day-of-week, and seasonal variability. The corresponding profiles are presented in Fig. 7.

Profiles for public transport and ship operations on Lake Zurich were deduced from published time schedules separately for weekdays, Saturdays and Sundays. Public transport is only a relevant source of  $CO_2$  in the outskirts because in the city center all busses (and trams) run on electric power. Shipping is mostly limited to the months of April to October. There are slightly more ship cruises during weekends. The profiles for industrial activities were taken from Kuenen et al. (2014).

Temporal variability of emissions from human respiration was simulated with two separate diurnal profiles, one for residents and one for employees. The profile for residents was deduced from a standardized building occupancy profile for residents from the Swiss Society of Engineers and Architects (black line in Fig. 7(a)). The profile for employees is simply a mirrored image of the profile for residents, assuming that persons being residents at night turn into employees during the day. The two opposing

**Figure 7.** Temporal scaling factors (normalized to 1) for the sectors industry, public transport, shipping and human respiration from residents and employees. (a) Diurnal cycle of hourly factors, (b) day-of-week cycle of daily factors, (c) seasonal cycle of monthly factors. There are three different diurnal cycles for public transport and shipping reflecting differences between weekdays (solid line), Saturdays (dotted) and Sundays (dashed).

profiles are not only applied to different spatial distributions of residents and employees (see Fig. 5), but also to different total numbers of persons in each category. According to the Quartieranalyse, the number of employees in Zurich is larger than the number of residents, because more people are commuting into the city than in the opposite direction. This leads to overall higher total emissions from human respiration during daytime. An additional reduction at night would be expected from lower respiration during sleeping than during daytime activities (Cai et al., 2022), but this was not accounted for. For both residents and employees we assumed that the maximum occupancy was only 80% of the registered numbers (i.e. of the maximum capacity) to account for the fact that neither are all residents always at home nor all employees always at their workplaces.

## 2.6.2 Vegetation photosynthesis and respiration



Temporal variability of vegetation photosynthesis and respiration depends on plant phenology and environmental conditions. As explained earlier, these processes were simulated with VPRM. The main inputs to VPRM are ambient 2 m temperature, solar radiation, and two satellite-based indices, the Enhanced Vegetation Index (EVI) as a measure of greenness, and the Land Surface Water Index (LSWI) as a measure of soil water availability. Using these inputs, VPRM computes GPP and RE using vegetation-specific parameters for the type of vegetation present in the domain, which in our case was deciduous trees, evergreen trees, mixed forest, grassland and cropland. The equations and parameters used here are provided in Stagakis et al. (2025).

Because the catalogue approach requires a uniform scaling of the fluxes from a given vegetation type, we had to use a single time series of temperature, radiation, EVI and LSWI for the whole city. As a result, all pixels covered by a certain vegetation type had identical fluxes and the corresponding  $CO_2$  concentration field calculated by GRAL was scaled uniformly by the ratio between the actual flux computed by VPRM and the unit flux. 2 m temperature and solar radiation were taken from the

**Figure 8.** Enhanced Vegetation Index from Sentinel-2 on 3 June 2023. The patches used for scaling the fluxes of four different vegetation types are overlaid in different colors.

site Zurich Kaserne, which is an urban background site of the Swiss national air pollution monitoring network NABEL. EVI and LSWI were computed from 10 m resolution Sentinel-2 satellite images acquired every few days. An example of EVI on a cloud-free day in summer 2023 is presented in Fig. 8, which shows high values over the forested areas, grass and croplands, and low values over the densely built-up areas and water surfaces. To compute typical time series of EVI and LSWI, the values of representative patches for each vegetation type were averaged. These patches are marked as colored rectangles in Fig. 8. Due to frequent cloud cover, the time series had to be gap-filled using a robust interpolation combining outlier removal and cubic interpolation. We compared our flux estimates using this simplified approach with the detailed simulation accounting for spatially varying temperature, radiation, EVI and LSWI fields presented in Stagakis et al. (2025) and found differences in annual mean fluxes of GPP and RE of 3% and 26%, respectively, and RMSEs of the spatially and temporally resolved differences of about 36% and 42% of the annual mean.

# 2.7 Catalogue entry selection


Once the GRAMM/GRAL catalogue is computed, a time series of concentration fields is quickly generated for any chosen period by selecting for each hour the entry best matching the actual meteorological situation and applying the temporal factors described above to scale the individual concentration fields. The selection of the best matching entry is primarily based on the comparison between simulated and observed winds at a selection of weather stations inside and around the city. We employed

the approach proposed by May et al. (2024), which eliminates the tendency of the method of Berchet et al. (2017b) to select entries with on average too low wind speeds. In essence, the method computes a score for each measurement hour and catalogue entry based on deviations between measured and simulated wind speeds and directions at all sites, excluding entries where the measured radiation is not compatible with the corresponding stability class (Berchet et al., 2017a). We selected 18 sites within the GRAL domain, which are those presented in Fig. 1 plus a few sites operated by the city of Zurich, MeteoSwiss, and ETH Zurich. The complete list of sites including statistics of the performance of the model is presented in Table 3. Not all sites were used for the matching procedure because they were not located on a rooftop or, as in the case of Uetliberg, on a high mountain. The simple approach of driving a GRAMM simulation with a single vertical profile seems not to be able to capture the actual contrast between the flow near the surface and the free troposphere. Winds are measured at Uetliberg at 1043 m above mean sea level (amsl) on top of a 187 m tall television tower on the highest mountain in Zurich, often above the planetary boundary layer. Hilltop sites at lower elevation including ETH Hönggerberg (540 m amsl), Fluntern (556 m amsl), or Gubrist Gipfel (615 m amsl), on the other hand, were included in the matching procedure. All mid-cost CO<sub>2</sub> measurements presented in this study were taken below these hilltop sites. To better capture actual wind profiles throughout the lower troposphere it would likely be beneficial to use the newly developed GRAMM-SCI model (Oettl and Veratti, 2021; Oettl, 2021), which allows nesting the model into ERA5 reanalysis fields of the European Centre for Medium-Range weather forecasts. However, driving with ERA5 winds would require a largely different approach to generating the catalogue.

A typical example of the comparison between simulated and observed winds is shown in Fig. 9 for the station Stauffacherstrasse-Werdplatz for the first three months in 2023. Both wind speeds and direction are generally matched very well illustrating the capability of the approach to accurately capture actual flow conditions.

An important drawback of the catalogue approach is that the corresponding wind field represents a stationary flow situation that does not account for variations in speed and direction associated with gusts (Ashcroft, 1994) or calmer periods within the hour. As a result, the dispersion of the emitted trace gas is underestimated, which is best visible when analyzing the plumes downwind of point sources such as Hagenholz. When determining the CO<sub>2</sub> concentration field for a given hour, we therefore did not take one single catalogue entry but averaged over the five best scoring entries. This produces smoother concentration time series with less extreme values, which proved to be in much better agreement with the observations.

## 2.8 Background concentrations





The CO<sub>2</sub> concentration simulated for a given hour is composed of the sum of the contributions from sources and sinks in the city plus a background, which represents the CO<sub>2</sub> transported into the city through the model domain boundaries (see Equation 1). As mentioned in Sect. 2.1, three background sites located in different directions from the city are available. We constructed a background CO<sub>2</sub> time series as a weighted mean of the concentrations at these three sites with larger weights assigned to stations located upstream. For each hour, the weights were computed from the difference between the actual wind direction and the direction in which the sites are located relative to the city center using a Gaussian function with a width (1 σ) of 30°.

Figure 10a shows daily afternoon mean (12-16 UTC, 13-17 local time in winter) CO<sub>2</sub> concentrations at the three sites together with the weighted mean background for the two years of simulations. Here and in the following, we mostly limit the

**Table 3.** Comparison of simulated and observed winds: mean observed wind speed, root mean squared error (RMSE), mean bias (MB) and correlation (corr.) for wind speed and/or wind direction. The last column (match2obs) indicates whether the site was used for the catalogue entry selection.

| Site  | Full name                         | Obs. speed  | RMSE speed  | MB speed    | corr. speed | RMSE dir. | MB dir. | match2obs |
|-------|-----------------------------------|-------------|-------------|-------------|-------------|-----------|---------|-----------|
|       |                                   | $(ms^{-1})$ | $(ms^{-1})$ | $(ms^{-1})$ | $(ms^{-1})$ | (°)       | (°)     |           |
| duesh | Dübendorf-Empa Schallhaus 2       | 2.24        | 0.84        | -0.12       | 0.89        | 64.85     | 44.49   | yes       |
| ethh  | ETH Messfeld Hönggerberg          | 1.51        | 0.82        | 0.18        | 0.81        | 50.76     | 33.20   | yes       |
| gub   | Gubrist Gipfel                    | 2.14        | 1.13        | 0.64        | 0.79        | 47.56     | 31.22   | yes       |
| hard  | Hardau II                         | 3.36        | 1.26        | -0.88       | 0.86        | 47.42     | 29.00   | yes       |
| reh   | Affoltern                         | 1.65        | 0.86        | -0.06       | 0.90        | 64.13     | 42.88   | yes       |
| sma   | Fluntern                          | 2.09        | 0.92        | 0.12        | 0.87        | 52.38     | 33.28   | yes       |
| ueb   | Uetliberg Fernsehturm             | 6.75        | 2.06        | -3.06       | 0.71        | 56.32     | 38.20   | -         |
| wspm  | Wasserschutzpolizei Mythenquai    | 2.16        | 1.47        | -1.85       | 0.47        | 93.46     | 78.73   | -         |
| wspt  | Wasserschutzpolizei Tiefenbrunnen | 1.18        | 1.29        | -1.08       | 0.72        | 101.11    | 86.82   | -         |
| zbas  | Badenerstrasse Farbhof            | 2.04        | 0.84        | -0.19       | 0.85        | 57.18     | 41.05   | yes       |
| zgub  | Güterbahnhof                      | 2.18        | 0.83        | -0.29       | 0.88        | 55.82     | 45.22   | yes       |
| zhab  | Albisgüetli                       | 2.11        | 0.98        | -0.63       | 0.77        | 74.09     | 55.60   | yes       |
| zhhf  | Kantonales Labor Zürich           | 1.76        | 0.74        | -0.20       | 0.85        | 60.44     | 42.63   | yes       |
| zhhm  | Hardturmstrasse Förrlibuck        | 2.39        | 0.91        | -0.40       | 0.88        | 58.00     | 44.52   | yes       |
| zhmi  | Schule Milchbuck                  | 3.18        | 1.01        | -0.60       | 0.88        | 51.93     | 37.02   | yes       |
| zhrgn | Rosengartenstrasse                | 0.54        | 0.65        | -0.34       | 0.70        | 75.43     | 60.92   | -         |
| zhsf  | Stauffacherstrasse Werdplatz      | 2.74        | 1.00        | -0.66       | 0.84        | 48.68     | 31.77   | yes       |
| zhui  | Universität Zürich Irchel         | 2.50        | 1.39        | -1.62       | 0.73        | 79.55     | 68.31   | -         |
| zhwh  | Wollishofen                       | 2.47        | 0.99        | -0.37       | 0.75        | 59.66     | 42.68   | yes       |
| zjho  | Limmattalstrasse Höngg            | 1.53        | 0.79        | -0.12       | 0.83        | 68.22     | 51.33   | yes       |
| zsch  | Schimmelstrasse                   | 0.80        | 0.56        | -0.01       | 0.70        | 84.76     | 68.39   | -         |
| zsta  | Stampfenbachstrasse               | 1.89        | 0.85        | -0.35       | 0.81        | 56.26     | 38.12   | -         |
| ztie  | Tiefenbrunnen Wildbachstrasse     | 2.93        | 1.20        | -0.87       | 0.84        | 62.41     | 45.66   | yes       |
| ztle  | Letzigraben Telefonzentrale       | 2.22        | 0.85        | -0.38       | 0.89        | 50.51     | 33.57   | yes       |
| zubv  | Bankenviertel Bleicherweg         | 2.12        | 0.92        | -0.58       | 0.80        | 54.25     | 38.42   | -         |
| zuepk | Zürich Kaserne (Polizeikaserne)   | 2.27        | 0.86        | -0.41       | 0.84        | 53.69     | 35.22   | yes       |
| mean  |                                   | 2.26        | 1.09        | -0.54       | 0.8         | 62.65     | 46.09   |           |

**Figure 9.** Comparison of hourly simulated wind speed and direction from the GRAL catalogue matching procedure with observations above rooftop at the site Stauffacherstrasse Werdplatz.

analysis to afternoon concentrations as the model struggles reproducing nighttime concentrations (see Sect. 3.2). Overall, the concentrations followed each other rather closely. The site Breite-Birchwil to the northeast of the city occasionally showed large deviations from the other sites, especially during winter and most prominently on a few days in December 2023 and late January 2024. Because the Swiss Plateau is located between two prominent mountain ridges, the Alps and the Jura, the large-scale flow around Zurich is usually channeled along a south-west to north-east axis (Oney et al., 2015). The most frequent situation is southwesterly to westerly winds, in which cases Lägern and Beromünster are located upstream of Zurich. In these situations, Breite-Birchwil is downstream of the city or the airport of Zurich, which explains some of these deviations. Breite-Birchwil is upstream only under so-called "Bise" conditions, which is another prominent weather situation in Switzerland featuring cool winds from an easterly to northeasterly direction. Pronounced Bise conditions occurred, for example, in late January and early March 2023. During these periods, CO<sub>2</sub> was significantly enhanced at all three sites, possibly due to the air masses having spent more time over Europe than in the case of westerly winds. At the same time, the differences between the three sites were rather small, of the order of 4 ppm, probably due to the strong winds. A more detailed analysis of background CO<sub>2</sub> levels over a two-years period and of the origin of high CO<sub>2</sub> episodes is presented in Grange et al. (2024).




Uncertainties in background levels will be crucial for the inverse estimation of CO<sub>2</sub> emissions. Figure 10b shows the differences between individual station pairs as a function of wind direction. Wind directions in which a given station is located perfectly upstream of Zurich are labelled. The frequency of occurrence of individual wind directions (grey line) shows two

Figure 10. (a) Time series of daily afternoon (12-16 UTC) mean  $CO_2$  concentrations at the three background sites Beromünster, Lägern and Breite-Birchwil and the constructed Zurich background time series. (b)  $CO_2$  difference between station pairs (in terms of  $1\sigma$  standard deviation) as a function of wind direction measured at Breite-Birchwil. The grey line is the frequency of occurrence per  $20^{\circ}$  wind direction bin.

distinct peaks for southwesterly to westerly and for northwesterly (Bise) directions. During southwesterly to westerly winds, Beromünster and Lägern are located upstream, with their concentrations agreeing within about 2.5 to 4.5 ppm. During Bise situations, Lägern and Beromünster showed average differences of about 4 ppm from the upstream Breite-Birchwil site. The largest differences were observed during southeasterly to southerly winds, but these situations were infrequent. As previously mentioned, Breite-Birchwil occasionally had much higher concentrations when winds blew from the southwest and the station was in the city's outflow. The lowest station-to-station differences of around 2–3 ppm occurred during northwesterly to northerly winds, but these situations were rare. These station-to-station differences provide an indication of the uncertainty of the background levels. Assuming that the true uncertainty is somewhat lower because the background is a weighted average of the three stations, uncertainties in the background are probably in the order of 1 to 3 ppm.

## 3 Results and Discussion



## 3.1 Mean CO<sub>2</sub> distributions

Maps of GRAMM/GRAL simulated CO<sub>2</sub> averaged over two full years (September 2022 to August 2024) are presented in Figures 11 and 12 for the sum of all sources and for the three largest sectors, road traffic, heating, and industry. Figure 11 shows CO<sub>2</sub> at the lowest model level of 2 m, whereas Figure 12 shows the 0-100 m column mean. At the lowest level, traffic CO<sub>2</sub> (Fig. 11b) is highest along the road network. Concentration hotspots are visible at large traffic junctions and tunnel portals. For the column-averages (Fig. 12b), the roads are still visible but the signal is strongly diluted.

**Figure 11.** Maps of CO<sub>2</sub> without background at 2 m above ground averaged over the two-year period September 2022 - August 2024 for total emissions and the three largest sectors. (a) All anthropogenic sources (all sectors in Table ?? except 14, 50, 51, 52, 53, 54, 61, 62, 71, 72, (b) road traffic (sectors 6, 7, 9, 34), (c) heating (sectors 11, 12, 35, 36), (d) industry (sectors 15, 17, 37, 74). White patches correspond to buildings taller than 2 m.

Heating (GNFR sector C in Table 2) is the largest source, accounting for 46% of total emissions. Compared to traffic, however, CO<sub>2</sub> concentrations from heating are significantly more homogeneous at the 2 m level (Fig. 11c). This is because heating emissions are released above roof level, allowing them to disperse extensively before reaching the surface. In the vertical column (Fig. 12c), several point sources stand out, which correspond to combined power and heat production plants.


CO<sub>2</sub> from industrial sources is largest outside the city borders with some exceptions (Fig. 11d). At 2 m above ground, several localized sources are visible inside the city, which correspond to small industries, such as breweries, bakeries and crematoria. Emissions from construction work are also comprised in this category, which explains the high concentrations along a highway in the Schwamendingen district in the northeastern part of the city, where major construction work is being carried out to place the highway into an enclosure by 2025. In the vertical column (Fig. 12d), the waste-to-energy plant Hagenholz stands out prominently. The radial pattern emerging from this source emphasizes the limitations of the catalogue approach producing

Figure 12. Same as Fig. 11 but for CO<sub>2</sub> averaged over the vertical column from 0-100 m above ground.

static CO<sub>2</sub> fields for selected wind directions. The pattern also highlights the main wind directions in the city; winds were predominantly from westerly, southerly and northeasterly directions, whereas easterly to south-easterly directions were rare.

## 3.2 Comparison to measured CO<sub>2</sub>


In this section, we present a first comparison of model simulations against observations from the mid-cost  $CO_2$  sensors. The comparison is focused on time series of daily afternoon values at one rooftop site (Wollishofen), one site in a large backyard (Zürich Kaserne) considered as an urban background site, and one street level site (Schimmelstrasse). Furthermore, mean diurnal cycles in different seasons averaged over all rooftop sites and Zürich Kaserne are presented. A more quantitative evaluation will be presented in a forthcoming study, where measurements and simulations will be combined in a Bayesian emission inversion framework.

The time series of daily afternoon (12-16 UTC, i.e. 13-17 local time in winter) mean CO<sub>2</sub> concentrations at Zurich Kaserne, 440 a station in the city center, is presented in Fig. 13. The top panel shows the comparison for two years from September 2022 to August 2024. The lower panel is a zoom into five months during the annual heating period, when anthropogenic emissions

Figure 13. Time series of measured and simulated daily afternoon (12-16 UTC) mean  $CO_2$  at the station Zurich Kaserne. (a) Full period of 2 years. (b) Zoom into five winter months from October 2023 to February 2024.

are highest and biospheric contributions smallest. The seasonal trend with a maximum in winter and a minimum in summer is dominated by variations in the background (gray line) that is estimated from the surrounding background stations. The concentrations observed inside the city (blue line) are generally higher than the background due to  $CO_2$  added by urban activities. This is also true during summer when the biosphere is most active, suggesting that anthropogenic emissions outweigh biospheric uptake in the city even during the growing season. This is consistent with the analysis presented in Stagakis et al. (2025), which suggested that anthropogenic fluxes in the central parts of the city are roughly three times larger than the biospheric fluxes during summer.




The simulated values (red line), which are composed of the background plus  $CO_2$  simulated by GRAMM/GRAL, closely track the observations both in terms of temporal variability and absolute values. As described by Grange et al. (2024), a significant fraction of the observed variability is explained by the background, which does not only vary seasonally but also on synoptic time scales with changing weather conditions. The  $CO_2$  emitted from the city adds to this variability, especially during winter, as highlighted in panel b. Periods of elevated concentrations such as the events around early December 2023 and late January 2024 are composed of both enhanced background  $CO_2$  and enhanced  $CO_2$  from the city. These periods are typically associated with stagnant high-pressure situations with low winds and shallow atmospheric boundary layers allowing  $CO_2$  to build up both regionally and locally inside the city. This illustrates the critical importance of tracking the variability in background concentrations. Daily afternoon  $CO_2$  concentrations in the city are on average about 10 ppm higher than the regional background, which is significantly larger than the uncertainty of the mid-cost sensors of about 1 ppm (Grange et al., 2024). In winter, these enhancements can reach up to 50 ppm.

Figure 14. Same as Fig. 13 but at station Wollishofen.

An example time series at a rooftop station is presented in Fig. 14. Wollishofen is the southernmost station of the mid-cost network near the southwestern city border (see Figure 1). The CO<sub>2</sub> variability at this site is very similar to that at Kaserne, with some events even exceeding the magnitude of the enhancements at Kaserne. In comparison to Kaserne, the model's performance is lower. While many of the observed pollution episodes are tracked well, some events are significantly underestimated. The performance at other sites is very comparable to the two sites presented here.

Figure 15 shows the time series of daily afternoon CO<sub>2</sub> at Schimmelstrasse, one of three street-level traffic sites. While the observed seasonal and day-to-day variability is generally well captured, there is a tendency to overestimate CO<sub>2</sub> at this site. An even greater tendency to overestimate CO<sub>2</sub> is evident at the Rosengartenstrasse traffic site, whereas at the third traffic site (Stampfenbachstrasse), the model's performance is comparable to that at rooftop sites (not shown). One likely reason for the overestimation at Schimmelstrasse and Rosengartenstrasse is the model's limited capability to resolve flow in street canyons.

Visual inspection of the model simulations suggests that wind speeds are underestimated in many street canyons due to limited resolution. As a result, CO<sub>2</sub> accumulates too strongly. Another possible factor is that the model does not represent turbulence induced by traffic. This is consistent with several modelling studies (e.g., Wang and Zhang, 2009) that have demonstrated the importance of vehicle-induced turbulence for predicting the spatial gradients of air pollutants near roadways.

While afternoon concentrations are well captured by the model, this is not true for other hours of the day. The seasonal mean diurnal cycle of observed and simulated  $CO_2$  averaged over all sites is presented in Fig. 16. In all seasons, the minimum  $CO_2$  concentration occurs in the afternoon due to the dilution of surface emissions within a deep atmospheric boundary layer. During spring and summer, uptake by vegetation photosynthesis adds to this minimum. While the shape of the diurnal variation is well captured by the model in all seasons, the amplitude is substantially underestimated.

Figure 15. Same as Fig. 13 but for the traffic site Schimmelstrasse.

Although the anthropogenic emissions are smallest in summer, the amplitude of the diurnal cycle is much larger compared to winter. One explanation is the stronger contrast between daytime and nighttime boundary layers in summer. Another reason is the diurnal cycle of biospheric fluxes with net emissions during nighttime and net uptake during daytime. The underestimation of the amplitude during summer could thus be related to an underestimation of the diurnal variability in net ecosystem exchange. The analysis presented in Stagakis et al. (2025) suggested that VPRM is in good agreement with other biospheric flux models and with measurements of soil and grass respiration, but that analysis was limited to urban parks and did not represent other types of vegetation such as street trees. A further explanation could be that the temporal scaling factors used in GRAMM/GRAL (see Fig. 6 and 7) do not accurately represent the contrast between daytime and nighttime emissions. The underestimation of nighttime CO<sub>2</sub> by the model suggests that nighttime emissions are too low. However, this may only be true for heating, since the traffic profile is based on actual traffic counts.

We also investigated whether the static  $CO_2$  distributions of the catalogue account for the potential accumulation of  $CO_2$  over multiple hours. Such accumulation is expected to be particularly important under stable low-wind situations. To test this, we performed a series of dynamic simulations over 14 hours for one arbitrarily selected source group and for different stability classes and wind speeds. These simulations are identical to those performed when building the catalogue, but the Lagrangian particles carrying the emitted  $CO_2$  are followed over the full time of the simulation or until they leave the domain. In these dynamic simulations,  $CO_2$  integrated over the whole domain was initially zero and then increased with time until it reached a steady-state equilibrium between emissions and loss through transport out of the domain. The equilibrium was reached after about 7 hours in case of an extremely stable situation (stability class G) with low winds  $(0.25 \text{ m s}^{-1})$  but already after less than 2 hours in case of a neutral situation (stability class D) with stronger winds  $(4 \text{ m s}^{-1})$ . These equilibrium levels were almost

Figure 16. Seasonal mean diurnal cycle of  $CO_2$  averaged over all rooftop-level mid-cost stations excluding the site zjho. The blue shaded area denotes the standard deviation of the hourly values.

identical to the corresponding catalogue entries, confirming that accumulation over multiple hours is fully accounted for by the static solutions of the catalogue.





Another possible explanation is inaccurate vertical mixing in the model at night caused by an inaccurate representation of atmospheric stability. To investigate this hypothesis, Figure 17 presents seasonal mean diurnal cycles of the frequency distribution of stability and simulated wind speeds averaged over all sites. The figure suggests that the model, or more precisely the catalogue entry selection procedure described in Sect. 2.7, realistically represents the diurnal cycle of stability with neutral to unstable conditions (classes D to A) dominating during daytime and neutral to stable (classes D to G) during night. Very stable (F and G) and very unstable (A and B) conditions occured more frequently in spring and summer than in winter, which is consistent with the larger dynamics in atmospheric boundary layer depth and surface heat exchange during summer (Edwards et al., 2011; Brunner et al., 2015). There is thus no indication for a major misrepresentation of the atmospheric boundary layer dynamics by the model. Nevertheless, certain stability classes may be under- or overrepresented, which can have significant consequences. As demonstrated by Ars et al. (2017) and Hanfland et al. (2022), a shift by only one stability class can lead to differences in peak concentration and volume of a plume from an individual emission source by up to a factor of two. Further analyses will be needed to better understand the reasons for underestimating the amplitude in the CO<sub>2</sub> diurnal cycle and finally to mitigate this issue.

Overall, the catalogue approach applied to the GRAMM/GRAL model captures wind speeds and direction within the city with good accuracy and low bias as illustrated by Fig. 9 and Table 3, which is an important prerequisite for accurate simulation of  $CO_2$  dispersion in the city. This is illustrated by a recent study of Ponomarev et al. (submitted), using the mesoscale atmospheric transport model ICON-ART for estimating the  $CO_2$  emissions of the cities of Zurich and Paris, which showed that during periods where wind speeds were overestimated by ICON-ART, the model substantially underestimated the observed concentrations.

**Figure 17.** Seasonal mean diurnal cycle of the probability distribution of atmospheric stability and of the wind speed averaged over all sites. Atmospheric stability ranges from very unstable (A) to very stable (G), with D representing neutral conditions.

Future studies should investigate in more detail the main sources of model errors, which include: 1. Errors in representing the mean flow, especially wind speed, and its variability across the city. Our results suggest that these errors are comparatively small. Vertical wind profiles could be evaluated against Doppler wind lidar measurements, which were performed in Zurich during a limited period of time between September 2022 and March 2023. 2. Inaccuracies in turbulent mixing, notably due to errors in the stability class selection. A possible way forward could be to use the Eddy flux tower measurements at Hardau to determine atmospheric stability instead of letting the model select the stability class based on the catalogue selection procedure. 3. Errors due to the discretized nature of the catalogue approach. This was mitigated to some extent by averaging over the five best scoring situations rather than choosing a single one. 4. Errors due to the resolution not being sufficient to fully resolve the flow in street canyons. The increased deviations of the model from sensors at street level suggest that this is a significant limitation for such sites. 5. Model representation errors. At some sites, especially those at street level, we noticed that sampling the model output only one grid cell (10 m) away or one altitude level higher or lower, significantly altered the results. Capturing the concentrations at sites influenced by nearby sources will remain a challenge. Such sites will likely have to be excluded in inverse modeling. 6. Inaccuracies in background concentrations. Uncertainties in daily afternoon background levels are likely in the order of 1 to 3 ppm, which is significant compared to the average enhancements from emissions and biospheric fluxes in the city of about 10 ppm. 7. Sampling noise of the Lagrangian particle dispersion model. Since the concentration in a given volume is determined from the CO<sub>2</sub> mass of all Lagrangian particles in this volume, the statistical noise depends on the number of simulated particles. The selection of a suitable number of particles is a tradeoff between precision and computational cost. Although this noise may be significant at certain locations and times, it does not introduce systematic errors. 8. Errors due to applying the sector-specific temporal profiles uniformly across the city. These errors depend on the sector and may vary considerably over time. Examples of increased errors are a traffic congestion in one part of a city, or a temporal shutdown



of an industrial source. To enhance the flexibility of the model to capture such situations, it would be necessary to perform simulations not only per sector but also per location.

## 4 Conclusions








Observation-based estimates of urban  $CO_2$  emissions can help cities monitor their progress towards climate neutrality. However, there are many different monitoring strategies, ranging from measuring large-scale gradients between regions up- and downstream to quantifying the 'urban  $CO_2$  dome' with a network of sites within the city, and from measuring with a few high-precision instruments to measuring with a large number of low-cost sensors. The list of options also includes direct flux measurements using Eddy covariance (Stagakis et al., 2023), ground-based remote sensing (Dietrich et al., 2021), and airborne mass balance (Cambaliza et al., 2014). Furthermore, a given observation strategy needs to be supported by an appropriate modeling and emission estimation methodology.

In this study, we have presented a modeling framework tailored to support observations from a dense network of low- or mid-cost  $\mathrm{CO}_2$  sensors and suitable for cities with complex topography such as Zurich. The model is able to capture the mesoscale flow shaped by topography and land cover as well as the local flow around buildings, which is an advantage for representing gradients between different sensors that are more or less strongly influenced by nearby emissions. Instead of simulating the full temporal dynamics of meteorology and  $\mathrm{CO}_2$  dispersion, a catalogue of static situations is created, from which the dynamic evolution is reconstructed by selecting for a given hour the catalogue entry best matching the actual meteorology. Although producing the catalogue is computationally expensive, situation-matching and subsequent data processing are highly efficient, allowing the creation of concentration time series for an entire year within minutes.

Another advantage of the model is its ability to represent emissions at the highest possible level of spatial detail by treating them as either point, line or area sources. We have described the setup of the model for the city of Zurich, where we took advantage of the uniquely detailed inventory produced by the city administration. The CO<sub>2</sub> exchange was simulated with the highly parameterized but observation-driven Vegetation Photosynthesis and Respiration Model (VPRM), for which a detailed vegetation cover map was created down to the level of individual trees. The model was driven by measurements of ambient temperature and radiation, and by high-resolution satellite observations from Sentinel-2.

GRAMM/GRAL has been shown to capture very well the temporal variability and magnitude of the urban signal when compared against daily afternoon observations from the ZiCOS mid-cost CO<sub>2</sub> sensor network. This demonstrates the potential for a successful integration of observations and simulations for inverse emission estimation, which will be presented in a forthcoming study.

Despite the mature status of the model and the promising results, we have also identified a number of weaknesses that deserve more attention in the future. The first one is the limitations of the current catalogue approach. Instead of defining weather situations in a rigid parameter space of wind and stability conditions, it would likely be better to create a catalogue tailored to actually occurring weather situations. This would allow resolving the most frequent situations with more detail, for example with a finer spacing in wind directions, while removing situations that do not occur at all.

A second weakness is the inability to represent spatially inhomogeneous emission variations, since the same temporal scaling factors are applied uniformly across the city. To solve this problem, one could separate the CO<sub>2</sub> tracers not only by sector but also by region. This would significantly increase the number of tracers to be simulated, but it would have minimal impact on the computational cost because the number of Lagrangian particles to be followed would remain nearly unchanged.






A third weakness is the strong underestimation of  $CO_2$  accumulation in stable low-wind situations, which typically occur at night. The reasons for the underestimation are not yet understood but need further investigation.

The GRAMM/GRAL model can be set up for any city, provided that suitable input data sets are available. For example, GRAMM/GRAL has already been set up for  $CO_2$  simulations over the cities of Paris and Munich, where similar  $CO_2$  sensor networks have been established as in Zurich within the ICOS-Cities project.

The most critical input is the emission inventory, which is often not available or not of sufficient quality and detail. One way to address this is by downscaling an existing inventory using spatial proxies (Valencia et al., 2022) or by creating a new inventory from open data, as demonstrated for the city of Amsterdam (Mijling, 2020). Open source tools such as emiproc (Lionel et al., 2025), which was used to prepare the Zurich inventory for GRAMM/GRAL, or HERMES (Guevara et al., 2020) can support this. Other data sets such as 3-D buildings, land use and vegetation cover can usually be obtained from publicly accessible sources. For Europe, for example, high quality data sets such as the Urban Atlas Building Height 2012 or the new CLC+ Land Information System are provided through the Copernicus Earth observation programme. The main issue with most land cover products is that urban vegetation, especially street trees, is not represented with sufficient accuracy. For Zurich, additional vegetation observations from high-resolution aerial imagery and Lidar scanning were therefore included.

This study highlights the potential of high-resolution, building-resolving atmospheric modeling frameworks like GRAM-M/GRAL to accurately simulate urban  $\mathrm{CO}_2$  concentrations when paired with detailed emissions inventories and biospheric flux modeling. Its ability to support inverse modeling and provide near-real-time insights into emission patterns can greatly enhance transparency and accountability for cities pursuing net-zero goals, and it offers a scalable blueprint for other urban areas globally.

595 *Code availability.* The emiproc python tool used for preparing all emission inputs including temporal profiles is publicly available at https://doi.org/10.5281/zenodo.14614229

Data availability. The mid-cost sensor data used in this work are described and available via the ICOS Cities data portal (https://citydata.icos-cp.eu/portal). The hourly averaged observations have been made publicly accessible in a persistent data repository (Grange, 2024a, https://doi.org/10.5281/zenodo.13759332). With a total size of about 5 TB, the GRAMM/GRAL catalogue of 3D wind and CO<sub>2</sub> concentration fields is too large to be shared publicly. It is stored at the Swiss supercomputing center CSCS and can be made available by the authors upon reasonable request.

Author contributions. DB authored the article and prepared tables and figures. IS performed most of the simulations, contributed to figures and prepared a first draft of the manuscript. LB and LC contributed to data analysis and production of figures. LC prepared the emission inputs for GRAL. AB evaluated the VPRM results. SKG, PR and LE installed and maintained the measurement networks. JL and JC developed the land cover map for VPRM. All authors edited the paper text.

Competing interests. The authors declare that they have no conflict of interest.

605

610

Acknowledgements. The authors have received funding from ICOS Cities, a.k.a. the Pilot Applications in Urban Landscapes – Towards integrated city observatories for greenhouse gases (PAUL) project, from the European Union's Horizon 2020 research and innovation program under grant agreement no. 101037319. We would like to acknowledge the Swiss Supercomputing Center CSCS for access to the Piz Daint high-performance computer under project em05. We would like to express our gratitude to the departement Umwelt- und Gesundheitsschutz (UGZ) of the city of Zurich for making the detailed emissions inventory available to this project. The company Swisscom is acknowledged for supporting the mid-cost sensor measurements by providing access to antenna and space on various buildings within the city. AB was supported by the Swiss National Science Foundation under the Scientific Exchanges program (IZSEZ0\_22553). JC received funding from ERC consolidator grant CoSense4Climate (101089203).

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
