# Peer review of "Building-resolving simulations of anthropogenic and biospheric $CO_2$ in the city of Zurich with GRAMM/GRAL"

_EGUsphere, 2025_

## Author Comment (AC1)

**Reply to reviewer #1**

We would like to thank the reviewer for the critical reading and valuable suggestions. In the following, we address the points one by one presenting our replies in dark blue and changes to the manuscript in dark red.

This is an interesting and impactful paper. I recommend publication after attention to the comments below.

Major Comments

- The rationale for referencing prior work seems somewhat arbitrary. There are notable recent efforts with similar observations and similar modeling that are omitted (although not combined). A more thorough discussion of the recent literature would help the reader put this work in appropriate context.

The references covered prominent examples of urban $CO_2$ emission estimation studies, but we agree that recent studies employing similar observation and modeling approaches were missing. We added the following paragraph to the introduction section:

Networks of low- to mid-cost $CO_2$ sensors have already been deployed in other cities. A prominent example is the Berkeley Atmospheric $CO_2$ Observation Network (BEACO2N), a dense network of 35 nodes of $CO_2$ and air pollution sensors in the San Francisco Bay area (Shusterman et al., 2016), which was recently extended to other cities including Los Angeles (Kim et al., 2025), Glasgow, Providence and Heidelberg. Other examples are the Beijing–Tianjin–Hebei (JJJ) carbon monitoring system with 134 $CO_2$ sensor sites (Han et al., 2024) and a mid-cost $CO_2$ sensor network with eight sites in the city of Paris (Lian et al., 2024). In an inverse modeling study utilizing a mesoscale Lagrangian particle dispersion model, Turner et al. (2020) showed that the BEACO2N network successfully captured the reduction of $CO_2$ emissions during the COVID-19 pandemic.

Furthermore, we extended the sentence referencing GRAMM/GRAL studies at urban scale as follows:

Here, we describe one of these model systems, GRAMM/GRAL, which was previously used for air pollution simulations in the city of Zurich (Berchet 2017a,b) *and for optimizing the design of a $CO_2$ measurement network in the city of Heidelberg using an observing system simulation experiment (OSSE) (Vardag and Maiwald, 2024)*.

Furthermore, in response to a comment by reviewer #2 we added more information on the capability and limitations of a model like GRAMM/GRAL operating at 10 m resolution, where we cited additional urban modeling studies including CFD approaches (see our response to reviewer #2).

- It would be helpful to also include the spatial and temporal resolution of each input dataset in Table 1.

We added this information to Table 1. The resolution/accuracy was between 3 cm (land use cadastre of the city, vector data set) and 100 m (CORINE land cover). Regarding temporal resolution we added the following sentence:

Most data sets are static with update cycles between one and five years. When available, the reference year is included in the description. The two Sentinel-2 satellites together provide global coverage in five days. Europe is covered more frequently, but clouds lead to irregular sampling and corresponding data gaps of up to a few weeks, especially in winter.

- Why are there 3 peaks in heating/hot water demand diurnal profile? I understand this is based on a simulation from CESAR-P, but what is the behavioral reason, especially for the 3-6 am peak and then the 9-12 peak? Homes then offices? Surprising to me that it's not smoothed more in the morning

We agree that the profile didn't look very realistic. The time profile was based on a heating demand profile extracted from the CESAR-P simulation software. We had a closer look at the original publication of the Swiss Society of Engineers and Architects (SIA norm 385/1) and found much more realistic profiles of hot water demand for residential buildings for typical weekdays and weekend days (see Figure below). We implemented this new data set and regenerated all figures affected by the changed profile. The changed profile led to a small improvement in the representation of the seasonal mean diurnal $CO_2$ cycles in Figure 15 but had an almost negligible impact on Figures 13 and 14, which are based on afternoon mean concentrations.

[Figure]

- Additional quantification of the uncertainties in each aspect of the model would be helpful.

This is very challenging task (for any transport model) and in our view well beyond the scope of the present study. There is extensive literature on evaluation of the GRAMM/GRAL model system, which we are now citing in Section 2.2. as follows:

The performance of GRAMM/GAL in terms of representing meso- and microscale flows and pollutant dispersion has been evaluated extensively in previous studies. Using a similar

setup as in our study, May et al. (2024) investigated how well the flow in the complex topography of the city of Heidelberg, Germany, is represented by the model. They found very good agreement at 11 out of 15 measurement sites according to the performance criteria for mesoscale air quality models formulated by the European Environment Agency (EEA, 2011). At all 15 sites, the performance was within the extended benchmark criteria proposed by Oettl and Verati (2021) for the more challenging conditions in complex terrain. Furthermore, Oettl and Verati (2021) showed that winds simulated by GRAMM in Alpine topography are at least as accurate as those simulated by the numerical weather prediction model WRF at comparable resolution. The high quality of tracer dispersion simulated by the GRAL model has been demonstrated in several studies comparing model results with wind tunnel and tracer release experiments (Oettl, 2015) and with air pollutant measurements in street canyons (Oettl and Uhrner, 2011) and across a whole city (Berchet et al., 2017b).

It is also important to note that whenever a new version of GRAL is released, the model is evaluated against a large number of validation data sets (e.g. tracer release experiments) to demonstrate compliance with Austrian, German and other national guidelines for dispersion models. These tests are extensively described in the GRAL documentation available at https://gral.tugraz.at/download/documentations/

Furthermore, we added the following sentences at the end of the results section summarizing potential sources of model error:

Future studies should investigate in more detail the main sources of model errors, which include: (i) Errors in representing the mean flow, especially wind speed, and its variability across the city. Our results suggest that these errors are comparatively small. Vertical wind profiles could be evaluated against Doppler wind lidar measurements, which were performed in Zurich during a limited period of time between September 2022 and March 2023. (ii) Errors in turbulent mixing, notably due to errors in the stability class selection. A possible way forward could be to use the Eddy flux tower measurements at Hardau to determine atmospheric stability instead of letting the model select the stability class based on the catalogue selection procedure. (iii) Errors due to the discretized nature of the catalogue approach. This was mitigated to some extent by averaging over the five best scoring situations rather than choosing a single one. (iv) Errors due to the resolution not being sufficient to fully resolve the flow in street canyons. The increased deviations of the model from sensors at street level suggest that this is a significant limitation for such sites. (v) Model representation errors. At some sites, especially those at street level, we noticed that sampling the model output only one grid cell (10 m) away or one altitude level higher or lower, significantly altered the results. Capturing the concentrations at sites influenced by nearby sources will remain a challenge. Such sites will likely have to be excluded in inverse modeling. (vi) Errors in background concentrations. Uncertainties in daily afternoon background levels are in the order of 1 to 3 ppm, which is significant compared to the average enhancements from emissions and biospheric fluxes in the city of about 10 ppm. (vii) Sampling noise of the Lagrangian particle dispersion model. Since the concentration in a given volume is determined from the $CO_2$ mass of all Lagrangian particles in this volume, the statistical noise depends on the number of simulated particles. The selection of a suitable number of particles is a tradeoff between precision and computational cost. Although this noise may be significant at certain locations and times, it does not introduce systematic errors. (viii) Errors due to applying the sector-specific temporal profiles uniformly across the

city. These errors depend on the sector and may vary considerably over time. Examples of increased errors are a traffic congestion in one part of a city, or a temporal shutdown of an industrial source. To enhance the flexibility of the model to capture such situations, it would be necessary to perform simulations not only per sector but also per location.

- How should the reader thinking about the various sources of uncertainty in inversions. Presumably this approach largely aims to minimize the model representation error. Can we expect that other sources of error therefore are more important (measurement error, background error)?

Determining realistic uncertainties is crucial for atmospheric inversions. This includes uncertainties in the prior assumptions (emissions, background levels), the measurements, the atmospheric transport model, and the model representation. As this will be addressed in a forthcoming publication, we prefer not to discuss this here. The comparisons between the GRAMM/GRAL forward simulations and the observations presented in Figures 13, 14 and 15 provide an idea of the magnitude of the combined uncertainties. In response to reviewer #2, we added an analysis of the uncertainty associated with background concentrations, which is clearly more important than measurement uncertainty. The inversion will provide further insights into the separate contributions from prior assumptions versus measurement and model errors, but this is outside the scope of the present study.

- Additional discussion of whether a 10 m simulation is necessary for inverse modeling and offers significant benefit beyond lower resolution forward model would be helpful. The diurnal disagreement in Figure 15 is substantial. If this model is not replicating observed concentrations, would a lower resolution model have sufficed for inverse modeling? Some quantification of the improvement in modeled concentration of this model over others would be quite helpful or at least a discussion of how the reader might think about that question.

We have submitted an inverse modeling study using the mesoscale atmospheric transport model ICON-ART at 500 m resolution where we compared the simulations with the same mid-cost $CO_2$ observations (Ponomarev et al., submitted to Atmos. Chem. Phys.). The range of root-mean-square errors between ICON-ART and the rooftop sensors in Zurich was 10-16 ppm for afternoon mean concentrations. The corresponding values for GRAMM/GRAL are 8-12 ppm, suggesting that the higher resolution is indeed beneficial. However, these differences are not only due to resolution but also due to the largely different modelling approaches. Again, we prefer to discuss this in a forthcoming publication on inverse modeling with GRAMM/GRAL where we will analyze the performance statistics of GRAMM/GRAL in detail and discuss the comparison with ICON-ART.

The diurnal disagreement is indeed substantial, which will likely make it necessary to limit the data assimilation to those hours of the day, where the model performs best. Using only daytime or afternoon observations is common practice in atmospheric inverse modeling because it is much more difficult to reproduce mixing in shallow nocturnal boundary layers than well-mixed daytime atmospheric boundary layers.

- For lines 427 - 438 (and Figure 16): How does the seasonal and diurnal variability in the probability density of stability classes prove that the distribution of stability classes are correct?

We didn't claim that the distribution is correct, but made a more qualitative statement that there is "no indication for a major misrepresentation of the atmospheric boundary layer dynamics by the model." The diurnal cycles of the frequency distributions of stability classes in different seasons look plausible as argued in the manuscript, but it is possible that certain classes are over- or underrepresented. One disadvantage of dividing the weather situations into discrete stability classes is that the dispersion changes stepwise and significantly between neighbouring stability classes. As a result, even a small misattribution (shift by 1 stability class) can have significant consequences. We added the following sentence to this point:

Nevertheless, certain stability classes may be under- or overrepresented, which can have significant consequences. As demonstrated by Ars et al. (2017) and Hanfland et al. (2022), a small shift by only one stability class can lead to differences in peak concentrations and volumes of plumes from individual emission sources by up to a factor of two.

- I suggest adding an additional paragraph addressing the implications of this paper at the end.

The conclusions indeed ended too abruptly. We added the following lines at the end:

This study highlights the potential of high-resolution, building-resolving atmospheric modeling frameworks like GRAMM/GRAL to accurately simulate urban $CO_2$ concentrations when paired with detailed emissions inventories and biospheric flux modeling. Its ability to support inverse modeling and provide near-real-time insights into emission patterns can greatly enhance transparency and accountability for cities pursuing net-zero goals, and it offers a scalable blueprint for other urban areas globally.

Line-by-Line Edits

- Figure text is too small throughout the manuscript

We have increased the size of labels, legends and titles for all figures.

- What is meant by the last line of Table 1? The line with "Example" and "how we should do."  Final line is an "example" and should be removed.

Thank you, that was a mistake. The line should have been uncommented.

- Description Equation 1, for the unfamiliar reader, describing the units of each term would be helpful for understanding the equation

Concentrations are provided in ppm (dry air mole fraction) throughout the manuscript, since this is the units in which all measurements are reported. The temporal scaling factor tau has no units. We added this information.

- Figure 11, is the white in the city center missing data due to the building height higher than 2 m?

Yes, this is the reason. We added this information to the figure caption.

- Line 36: It is unclear what "suitable measurement" means.

We agree. The sentence was reformulated as follows:

Emissions from a city produce a gradient in atmospheric concentrations between regions upstream and downstream, which can be measured by a measurement network with stations suitably placed to capture these gradients.

- Figure 5: Color key shows values of "-999" to "0." The value -999 must be a filler for unknown values, so it should be removed or explained.

We added the following information to the figure caption:

White color indicates no information for the corresponding building block or zero occupancy.

- Figure 7: Images b and c require x axis labels.

We think that the labels ('Mon', 'Tue', etc. in Fig.7b and 'Jan', 'Feb', etc. in Fig.7c) together with the figure caption are sufficiently clear.

- Figure 8: How were the different vegetation patches selected as representative?

This was entirely based on visual inspection looking for areas with very homogeneous coverage by the corresponding vegetation (using information on vegetation type provided by the data sets listed in Table 2 as well as aerial satellite imagery).

- Table 3: How was the threshold determined for match2obs selected sites.

We did not apply a threshold to determine if a site was selected or not. As explained in the text, we excluded sites not located on rooftops and, as in the case of Uetliberg, on a high mountain. We also excluded the rooftop sites Zürich Irchel and Bankenviertel where the wind sensors were placed too close to the roof or building because of logistic constraints. We provide further information on the reason for excluding Uetliberg in a response to a comment of reviewer #2.

---

## Author Comment (AC2)

**Reply to reviewer #2**

This paper presents the GRAMM/GRAL model setup to simulate $CO_2$ concentrations in the city of Zurich. The manuscript is generally well-written and presents robust results, forming a solid basis for future studies. However, additional analysis and more in-depth discussion are necessary before the manuscript is ready for publication in *Atmospheric Chemistry and Physics*.

**Major Comments**

- **Line 330: Selection of sites for matching**

  Your claim is that the model can resolve flow within street canyons. Therefore, including stations located within street canyons for model validation should be appropriate. If including these stations deteriorates the model performance, this may indicate that GRAMM/GRAL does not adequately capture the street-level flow, raising questions about the justification for using it in such environments.

The idea of the matching procedure is to capture the mesoscale flow, which then forces the microscale flow including the flow around buildings. Even at the high resolution of our model, it is necessary to select sites measuring winds with sufficient spatial representativeness. It is true that we did not evaluate the quality of the microscale flow in this study. Unfortunately, the wind measurements in street canyons in Zurich (Stampfenbachstrasse, Rosengartenstrasse, Schimmelstrasse) are not suitable for this purpose. These measurements are taken at air quality stations a few meters away from busy roads and trees (Rosengartenstrasse, Schimmelstrasse) or very close to a building facade (Stampfenbachstrasse), situations that can not be represented by the model.

Although we didn't explicitly claim that we can resolve the flow in street canyons, this may have been concluded from our statement that "*the model operates at a high horizontal, building-resolving resolution 10 m x 10 m, which is required to capture the situation at the low-cost sensor sites at street level*". Since most street canyons in Zurich are no wider than a few tens of meters, a resolution of 10 m poses clear limitations to representing the flow in these canyons. A higher resolution (5 m or better) would be desirable, but so far has been prohibitive due to memory limitations (the model has only OpenMP but no MPI parallelization). We will add the following sentences in the introduction clarifying the capability and limitations of a model at this resolution:

The model operates at a high horizontal, building-resolving resolution, which is required to capture the situation at the low-cost sensor sites at street level, but it may also be an advantage to represent the $CO_2$ concentrations at rooftop level. Due to computational constraints, the horizontal resolution was limited to 10 m, which is sufficient to capture the flow in wide streets and open spaces such as parks, squares and rivers but it poses limitations for resolving the flow in most street canyons in Zurich, which are no wider than a few tens of meters. Despite these limitations, the resolution is orders of magnitude better compared to atmospheric transport models used in previous urban $CO_2$ studies, which typically operated at resolutions of one kilometer or coarser (Staufer et al., 2016; Feng et al., 2016; Turner et al., 2020; Lauvaux et al., 2020; Nalini et al., 2022). Computational fluid

dynamics models such as OpenFoam (Kubilay et al., 2018) and PALM (Maronga et al., 2020) have also been run over cities at building-resolving scale down to sub-meter resolution, but these simulations were limited to individual districts or short time periods due to the excessive computational costs.

Furthermore, we added a figure (new Fig. 15) comparing simulated $CO_2$ with one of the traffic sites. The site Schimmelstrasse was selected for this, as the model's performance is in between the other two traffic sites Stampfenbachstrasse and Rosengartenstrasse. We added the following text:

Figure 15 shows the time series of daily afternoon $CO_2$ at Schimmelstrasse, one of three street-level traffic sites. While the observed seasonal and day-to-day variability is generally well captured, there is a tendency to overestimate $CO_2$ at this site. An even greater tendency to overestimate $CO_2$ is evident at the Rosengartenstrasse traffic site, whereas at the third traffic site (Stampfenbachstrasse), the model's performance is comparable to that at rooftop sites (not shown). One likely reason for the overestimation at Schimmelstrasse and Rosengartenstrasse is the model's limited capability to resolve flow in street canyons. Visual inspection of the model simulations suggests that wind speeds are underestimated in many street canyons due to limited resolution. As a result, $CO_2$ accumulates too strongly. Another possible factor is that the model does not represent turbulence induced by traffic. This is consistent with several modelling studies (e.g. Wang and Zhang, 2009) that have demonstrated the importance of vehicle-induced turbulence for predicting the spatial gradients of air pollutants near roadways.

- **Line 330: Omission of the mountain station**

  Excluding the mountain station is a limitation, especially since the manuscript claims GRAMM/GRAL performs particularly well at such sites. While it is true that models often struggle to represent flow at mountain tops, this issue warrants discussion. Additionally, excluding higher elevation sites may point to problems in representing the vertical concentration profile. Please address these points in the revised version.

We agree that this deserves further discussion. We've added the following explanation:

The simple approach of driving a GRAMM simulation with a single vertical profile seems not to be able to capture the actual contrast between the flow near the surface and the free troposphere. Winds are measured at Uetliberg at 1043 m above mean sea level (amsl) on top of a 187 m tall television tower on the highest mountain in Zurich, often above the planetary boundary layer. Hilltop sites at lower elevation including ETH Hönggerberg (540 m amsl), Fluntern (556 m amsl), or Gubrist Gipfel (615 m amsl), on the other hand, were included in the matching procedure. All mid-cost $CO_2$ measurements presented in this study were collected below these hilltop sites. To better capture actual wind profiles throughout the lower troposphere, it would likely be beneficial to use the newly developed GRAMM-SCI model (Oettl and Verati, 2021; Oettl 2021), which allows nesting the model into ERA5 reanalysis fields of the European Centre for Medium-Range Weather Forecasts. However, driving with ERA5 winds would require a largely different approach to generating the catalogue.

- **Section 2.8: Background concentration**

  Please discuss how the choice of background concentration influences the results. Can you provide an uncertainty estimate, perhaps based on the spread among background stations? This is particularly important given your observation that a substantial portion of the concentration signal and its variability originates from the background. Moreover, this discussion is essential groundwork for any future inverse studies.

Uncertainties in the background will indeed be a critical aspect for the inverse modeling. We have revised and extended the discussion of background concentrations in two ways: First, instead of showing hourly values, we now present daily afternoon means but for the full 2 years instead of only a few months. Second, we added a panel b to Figure 10 showing pairwise differences between the background stations as a function of wind direction. To the first point, the paragraph was changed to

Figure 10a shows daily afternoon mean (12-16 UTC, 13-17 local time in winter) $CO_2$ concentrations at the three sites together with the weighted mean background for the two years of simulation. Here and in the following we mostly limit the analysis to afternoon concentrations as the model struggles reproducing nighttime concentrations (see Sect. 3.2). Overall, the concentrations followed each other rather closely. The site Breite-Birchwil to the northeast of the city occasionally showed large deviations from the other sites, especially during winter and most prominently on a few days in December 2023 and late January 2024.

To the second point, the following paragraph was added. The last sentence summarizes the expected uncertainty in background concentrations.

Uncertainties in background levels will be crucial for the inverse estimation of $CO_2$ emissions. Figure 10b shows the differences between individual station pairs as a function of wind direction. Wind directions in which a given station is located perfectly upstream of Zurich are labelled. The frequency of occurrence of individual wind directions (grey line) shows two distinct peaks for southwesterly to westerly and for northwesterly (Bise) directions. During southwesterly to westerly winds, Beromünster and Lägern are located upstream, with their concentrations agreeing within about 2.5 to 4.5 ppm. During Bise situations, Lägern and Beromünster showed average differences of about 4 ppm from the upstream Breite-Birchwil site. The largest differences were observed during southeasterly to southerly winds, but these situations were infrequent. As previously mentioned, Breite-Birchwil occasionally had much higher concentrations when winds blew from the southwest and the station was in the city's outflow. The lowest station-to-station differences of around 2–3 ppm occurred during northwesterly to northerly winds, but these situations were rare. These station-to-station differences provide an indication of the uncertainty of the background levels. Assuming that the true uncertainty is somewhat lower because the background is a weighted average of the three stations, uncertainties in the background are probably in the order of 1 to 3 ppm.

- **Line 384: Limited presentation of data**

  Since this paper aims to demonstrate the model's capabilities (and is not an inverse

study relying only on afternoon values), presenting only the mean diurnal cycle and two stations during the afternoon is insufficient. I recommend showing the full time series for the two selected stations and then arguing why a focus on afternoon values reduces mismatches. Otherwise, the comparison may be misleading and overly optimistic.

We added the full time series to the appendix. Note that the mean diurnal cycles also show the range in terms of +/- 1 standard deviation of the hourly values. We think this a clearer way of presenting the agreement (or disagreement) between simulations and observations than showing a busy time series of hourly values, where it becomes difficult to see the details.

- **Figure 15: Large discrepancies in diurnal cycles**

  The differences between modeled and observed diurnal cycles are substantial. While you mention possible causes such as incorrect VPRM input, flawed scaling factors, or PBLH errors, the discussion remains superficial. Given Zurich's extensive observational infrastructure, these discrepancies should be examined in more detail—for example, through comparison with other top-down estimates or vertical mixing data from tall towers. The discussion of catalog probabilities is weak and inconclusive; consider moving this part to the appendix and replacing it with a more thorough analysis of the discrepancies.

The discussion of the potential reasons for the substantial differences in the diurnal cycles is indeed somewhat unsatisfactory. We tested many hypotheses but, unfortunately, were unable to identify one major driving factor. One of the hypotheses was that the static hourly solutions of the catalogue do not properly account for the accumulation of $CO_2$ over multiple hours, which is expected to be particularly important under stable low-wind conditions. See our reply to the corresponding point below. Another possibility that we mentioned is too strong mixing at night, possibly due to the selection of not sufficiently stable situations. In response to reviewer #1, we added a discussion of the impact of selecting a wrong situation. Finally, background concentrations deduced from stations dozens of kilometers away may not be appropriate for low wind speed situations. Another approach for stable night-time conditions with low winds could be to use the concentrations in the city in the preceding afternoon as background. However, applying this idea would require substantial additional work as it will have to be tested under which situations this is useful and how the method could be blended with the existing background method, for example depending on wind speed and stability.

The discussion of the discrepancies in the mean diurnal cycles was thus expanded with a discussion of the dynamic simulations and with the potential impact of selecting a wrong stability class. We feel that the paper is already very long and that additional discussions would dilute the overall message. We will certainly continue investigating this aspect, for example following the above-mentioned idea of using afternoon concentrations as background for nights with stable low-wind conditions.

- **Line 475: Dynamic $CO_2$ simulation**

You refer here to simulating $CO_2$ in a dynamic manner, but this concept is introduced rather abruptly in the discussion section without being presented in the results. Please clarify and provide context. This aspect deserves better integration into the manuscript.

We agree that this was not explained in sufficient detail. GRAMM/GRAL allows simulating in dynamic mode, i.e. Lagrangian particles are transported in a dynamically changing flow field. Because of the strong underestimation of $CO_2$ in stable situations at night, we suspected that the static approach may not properly represent the accumulation of $CO_2$ over many hours. To check this, we performed simulations for a few selected situations in dynamic mode (but keeping the same situation over the simulation). The concentrations in the dynamic simulations increased with time until they reached a steady state, which almost exactly matched the concentration levels of the static simulation for the same situations. This convinced us that the catalogue of static situations was built in a way that fully accounts for the buildup of $CO_2$ over multiple hours.

As an example, the figure below compares the results between transient (solid lines) and static simulations from the catalogue (dashed lines) for three situations with different stability and wind speed. The figure shows domain-integrated mass of $CO_2$ emitted by one arbitrarily selected source group. It shows that the transient simulations reach a constant level very closely matching the results from the static simulations. As expected, it takes longer to reach the steady state for an extremely stable situation with low winds of 0.25 m/s (situation 200, green) than for a neutral situation with stronger winds of 4 m/s (situation 921).

[Figure]

We moved the discussion of these dynamic simulations from the conclusions to the discussion of the comparison of the seasonal mean diurnal cycles in Section 3.2 and expanded it as follows:

We also investigated whether the static $CO_2$ distributions of the catalogue account for the potential accumulation of $CO_2$ over multiple hours. Such accumulation is

expected to be particularly important under stable low-wind situations. To test this, we performed a series of dynamic simulations over 14 hours for one arbitrarily selected source group and for different stability classes and wind speeds. These simulations are identical to those performed when building the catalogue, but the Lagrangian particles carrying the emitted $CO_2$ are followed over the full time of the simulation or until they leave the domain. In these dynamic simulations, $CO_2$ integrated over the whole domain was initially zero and then increased with time until it reached a steady-state equilibrium between emissions and loss through transport out of the domain. The equilibrium was reached after about 7 hours in case of an extremely stable situation (stability class G) with low winds (0.25 ,m s$^{-1}$) but already after less than 2 hours in case of a neutral situation (stability class D) with stronger winds (4 m s$^{-1}$). These equilibrium levels were almost identical to the corresponding catalogue entries, confirming that accumulation over multiple hours is fully accounted for by the static solutions of the catalogue.

- **Line 479: Potential for further investigation**

  As mentioned earlier, there are additional data and modeling resources that could help investigate the discrepancies observed in this study. It would strengthen the manuscript to make use of these tools or at least outline how they could be used in follow-up work.

There is indeed additional data that could help investigate the model's performance. As mentioned in the manuscript, we made use of ecosystem observations collected in Zurich to evaluate the performance of the VPRM model (see Stagakis et al., 2025; https://bg.copernicus.org/articles/22/2133/2025/). This analysis showed comparable performance to other vegetation models, but the analysis was limited to urban parks and gross photosynthetic production (GPP) could only be evaluated qualitatively. Additional work is in progress: Li et al. have just submitted a publication comparing different VPRM versions and evaluating them against ecosystem observations in both Zurich and Munich and including a quantitative evaluation of GPP. Still, one major drawback is that so far no observations are available for street trees. Such observations are currently being collected in the framework of a Swiss research project.

Another potentially useful data set is vertical wind profiles from two Doppler lidars deployed in Zurich during about 7 months. The first lidar (Metek Windranger 200) probed the lowest 100 m above ground, the second (Leosphere Wind Cube 100) higher altitudes starting at 200 m. Unfortunately, the profiles of the Windranger were strongly influenced by nearby high-rise buildings, but the measurements at the highest level (100 m) agreed very closely with the in-situ measurements at Hardau II. Since the latter were included in the catalogue matching procedure, the measurements from the Windranger do not add much information. Profiles from the second lidar measuring above 200 m, on the other hand, could be useful to evaluate the GRAMM/GRAL wind profiles. Since the lidar was deployed for only a few months, its measurements could not be integrated into the catalogue selection procedure.

A third potentially useful data set is the Eddy covariance measurements at the Hardau tower. Some of these measurements have been described by Hilland et al. (2025, under review), a publication now cited in our manuscript. These measurements could potentially be used to

evaluate the diurnal cycles of the probability distribution of stability classes presented in Figure 16 (now Fig. 17). However, flux measurements performed on top of a tall building at 112 m above ground in a heterogenous urban area need to be treated with great care, e.g., to account for issues such as vertical decoupling during stable situations, storage fluxes, distortion of the flow by the building, etc.  There is not enough space in this manuscript to describe such a complex data set and how it could be applied to determine stability classes. Further studies using the Eddy covariance observations are currently in preparation, which will serve as a much better basis and reference for future analyses.

Instead of providing additional analyses, we added a paragraph at the end of the results section describing different potential sources of error and how additional observations could help address them in follow-up studies. Please see our response to reviewer #1 asking for "additional quantification of the uncertainties in each aspect of the model".

**Minor Comments**

- **Line 58:** Please clarify what "As an alternative" refers to, or consider removing the phrase for clarity.

Thank you, this was not clear. The sentence was reformulated as follows:

To enable an alternative method for quantifying the $CO_2$ emissions of the city, an Eddy covariance system for direct $CO_2$ flux measurements was installed on a 16.5 m mast on top of one of the tallest (95.3 m) buildings in the city. These measurements are presented in Hilland et al. (2025) and were not used in this study.

- **Line 208ff:** Remove the brackets for Glauch et al. 2025.

The reference was changed as suggested.

- **Table 1, last line:** Please delete this line.

This was a mistake also pointed out by reviewer #1. The line was deleted.

- **Line 251:** The sentence "Possible differences … cannot be accounted for in this way" is not entirely accurate. In principle, these differences could have been addressed by distinguishing between inner-city and outer emissions through separate source groups.

This is true. We added
.. unless they are simulated as separate categories.

- **Line 306:** The statement "requires a uniform scaling of the fluxes from a given vegetation type" suggests a limitation, but this could be addressed with a more refined approach.

This is indeed a limitation and we don't think it can easily be overcome, unless vegetation in different parts of the city (or different altitudes) is treated as separate tracers.

- **Line 358:** "Rather small gradients" is vague. Please provide a quantitative value or range.

They were of the order of 4 ppm. We will add this information. Note that the section on the background has been revised significantly.

- **Line 461:** It would be appropriate here to acknowledge that GRAMM/GRAL struggles to reproduce dynamics outside of afternoon hours.

These difficulties are mentioned a few lines later.

- **Literature:** Please refer To Vardag and Maiwald (2024), who have already applied GRAMM/GRAL for inversion studies.

Thank you, this reference was badly missing. We already added it in response to a request of reviewer #1 to include more recent relevant literature in the introduction (see corresponding response to reviewer #1).